# Cohesin-independent STAG proteins interact with RNA and R-loops and promote complex loading

Hayley Porter[1†], Yang Li[1†], Maria Victoria Neguembor[2], Manuel Beltran[3], Wazeer Varsally[1], Laura Martin[2], Manuel Tavares Cornejo[3], Dubravka Pezić[1], Amandeep Bhamra[4], Silvia Surinova[4], Richard G Jenner[3], Maria Pia Cosma[2,5,6], Suzana Hadjur[1]*

[1]Research Department of Cancer Biology, Cancer Institute, University College London, London, United Kingdom; [2]Centre for Genomic Regulation (CRG), Barcelona Institute of Science and Technology, Barcelona, Spain; [3]Regulatory Genomics Group, Cancer Institute, University College London, London, United Kingdom; [4]Proteomics Research Translational Technology Platform, Cancer Institute, University College London, London, United Kingdom; [5]Universitat Pompeu Fabra (UPF), Barcelona, Spain; [6]Institució Catalana de Recerca i Estudis Avançats (ICREA), Barcelona, Spain

**Abstract** Most studies of cohesin function consider the Stromalin Antigen (STAG/SA) proteins as core complex members given their ubiquitous interaction with the cohesin ring. Here, we provide functional data to support the notion that the SA subunit is not a mere passenger in this structure, but instead plays a key role in the localization of cohesin to diverse biological processes and promotes loading of the complex at these sites. We show that in cells acutely depleted for RAD21, SA proteins remain bound to chromatin, cluster in 3D and interact with CTCF, as well as with a wide range of RNA binding proteins involved in multiple RNA processing mechanisms. Accordingly, SA proteins interact with RNA, and R-loops, even in the absence of cohesin. Our results place SA1 on chromatin upstream of the cohesin ring and reveal a role for SA1 in cohesin loading which is independent of NIPBL, the canonical cohesin loader. We propose that SA1 takes advantage of structural R-loop platforms to link cohesin loading and chromatin structure with diverse functions. Since SA proteins are pan-cancer targets, and R-loops play an increasingly prevalent role in cancer biology, our results have important implications for the mechanistic understanding of SA proteins in cancer and disease.

*For correspondence:
s.hadjur@ucl.ac.uk

[†]These authors contributed equally to this work

Competing interest: The authors declare that no competing interests exist.

## Editor's evaluation

This study reports that the Stromalin Antigen (SA) proteins play a key role in the localization of the cohesin complex and promote loading of the complex. It shows that SA proteins interact with RNA, RNA binding proteins and R-loops, even in the absence of cohesin, providing evidence for a role for SA1 in cohesin loading which is independent of the canonical cohesin loader NIPBL. The study open new perspectives to understand the links between cohesin loading and chromatin structure that would rely on R-loops.

## Introduction

Cohesin complexes are master regulators of chromosome structure in interphase and mitosis. Accordingly, mutations of cohesin subunits lead to changes in cellular identity, both during development and

in cancer (*Leiserson et al., 2015*; *Horsfield et al., 2007*; *Viny and Levine, 2018*). A prevailing model is that cohesin contributes to cell identity changes in large part by dynamically regulating 3D genome organization and mediating communication between distal regulatory elements (*Hadjur et al., 2009*; *Sofueva et al., 2013*; *Zuin et al., 2014*; *Kojic et al., 2018*; *Rao et al., 2017*; *Wutz et al., 2017*; *Fudenberg et al., 2018*). However, molecular insight into how and when cohesin subunits become associated with chromatin and contribute to this function in vivo in human cells is still lacking.

Most studies of cohesin function consider the Stromalin Antigen (STAG/SA) proteins as core complex members given their ubiquitous interaction with the tripartite cohesin ring (composed of SMC1, SMC3, and SCC1/RAD21). Rarely is the SA subunit considered for its roles independent of the cohesin ring, even though it is the subunit most commonly mutated across a wide spectrum of cancers (*Leiserson et al., 2015*; *Balbás-Martínez et al., 2013*; *Solomon et al., 2013*).

SA proteins contribute to cohesin's association with DNA (*Murayama and Uhlmann, 2014*; *Orgil et al., 2015*). The yeast SA orthologue is critical for efficient association of cohesin with DNA and its ATPase activation (*Murayama and Uhlmann, 2014*; *Orgil et al., 2015*). Separating interactions into SA-loader and cohesin ring-loader sub-complexes still impairs cohesin loading, indicating that SA functions as more than just a bridge protein (*Orgil et al., 2015*). Crystallization studies reveal a striking similarity of SA with NIPBL [the canonical cohesin loader (*Ciosk et al., 2000*)], in that both are highly bent, HEAT-repeat containing proteins (*Kikuchi et al., 2016*; *Hara et al., 2014*). Of note, NIPBL and SA interact together and wrap around both the cohesin ring and DNA to position and entrap DNA (*Li et al., 2018*; *Shi et al., 2020*; *Higashi et al., 2020*), implying a potential role for SA in the initial recruitment of cohesin to DNA alongside NIPBL. Further, SA proteins bridge the interaction between cohesin and CTCF (*Li et al., 2018*; *Xiao et al., 2011*; *Saldaña-Meyer et al., 2014*), and also bridge interactions with specific nucleic acid structures in vitro (*Bisht et al., 2013*; *Lin et al., 2016*).

Mammalian cells express multiple SA paralogs. SA1 binds to AT-rich telomeric sequences (*Bisht et al., 2013*; *Lin et al., 2016*) and SA2 displays sequence-independent affinity for particular DNA structures commonly found at sites of repair, recombination, and replication (*Countryman et al., 2018*). Consistent with this, results in yeast implicate non-canonical DNA structures in cohesin loading in S-phase. In vitro experiments show that cohesin captures the second strand of DNA via a single-strand intermediate (*Murayama et al., 2018*), and chromatid cohesion is impaired by de-stabilization of single-strand DNA intermediates during replication (*Zheng et al., 2018*). Together, these suggest that SA proteins and DNA structures may play a regulatory role in guiding or stabilizing cohesin localizations.

During transcription, the elongating nascent RNA can hybridize to the template strand of the DNA and form an R-loop, which is an intermediate RNA:DNA hybrid conformation with a displaced single strand of DNA (*García-Muse and Aguilera, 2019*). A multitude of processes have been linked to R-loop stability and metabolism. For example, co-transcriptional RNA processing, splicing, and messenger ribonucleoprotein (mRNP) assembly counteract R-loop formation (*Crossley et al., 2019*; *Li and Manley, 2005*). R-loop structures have also been shown to regulate transcription of mRNA by recruitment of transcription factors, displacement of nucleosomes, and preservation of open chromatin (*Boque-Sastre et al., 2015*; *Powell et al., 2013*). Hence, like at the replication fork, sites of active transcription accumulate non-canonical nucleic acid structures.

We set out to investigate the nature of SA proteins and cohesin loading to DNA. We discovered independent functions of the SA proteins, providing critical insight into the importance they play in their own right to direct cohesin's localization and loading to chromatin. In cells acutely depleted of RAD21, SA proteins remain associated with chromatin and CTCF where they are enriched at chromatin sites clustered in 3D. Moreover, we identify numerous, diverse cohesin-independent SA1 interactors involved in RNA processing, ribosome biogenesis, and translation. Consistent with this, SA1 and SA2 interact with RNA and non-canonical nucleic acid structures in the form of R-loops. Importantly, SA proteins are required for loading of cohesin to chromatin in cells deficient for NIPBL. Our results highlight a central role for SA proteins in cohesin biology and the cohesin-independent interaction of SA proteins with RNA processing factors opens up a new understanding of how SA dysregulation can impact disease development that moves us beyond the control of chromatin topology for gene expression regulation.

## Results

## SA interacts with CTCF on chromatin in the absence of the cohesin trimer

To determine how CTCF and cohesin assemble on chromatin, we used previously described (*Natsume et al., 2016*) human HCT116 cells engineered to carry a miniAID tag (mAID) fused to monomeric Clover (mClover) at the endogenous RAD21 locus and *OsTIR1* under the control of CMV (herein RAD21^mAC). RAD21^mAC cells were cultured in control EtOH conditions (EtOH) or in the presence of auxin (IAA) to induce rapid RAD21 degradation (*Figure 1—figure supplement 1a, b*). Immunofluorescence (IF) was used to monitor the levels of mClover, SA1, SA2, and CTCF (*Figure 1a and b*, and *Figure 1—figure supplement 1b*). As reported *Natsume et al., 2016*, acute IAA treatment dramatically reduced mClover levels compared to control cells (mean fluorescence intensity (MFI) reduction of 82%, p=2.9E-239). SA proteins and CTCF were also all significantly reduced, however the extent of the change was notably different. We observed a small but significant reduction in CTCF signal upon IAA treatment (mean reduction of 16%, p=1.6E-27). This was similar to the mean SA1 signal which was reduced by 22% compared to EtOH control (p=3.5E-21). However, SA2 levels mirrored more closely the effect on mClover, being reduced by 63% (p=1.9E-186), but not completely lost (*Figure 1b*). The retention of SA proteins despite the degradation of RAD21 was surprising given the fact that they are considered to be part of a stable biochemical complex.

We sought to validate these observations using an orthogonal technique and to establish whether the residual SA proteins retained the capacity to directly interact with CTCF. We prepared chromatin extracts from RAD21^mAC cells treated with EtOH or IAA and performed chromatin co-immunoprecipitation (coIP) to probe the interactions between SA proteins, RAD21 and CTCF. Both SA1 and SA2 interacted with RAD21 and CTCF in control cells as expected (*Parelho et al., 2008*; *Wendt et al., 2008*), with notable differences in their preferred interactions whereby SA2 more strongly enriched RAD21 while the SA1-CTCF interaction was significantly stronger than SA2-CTCF (*Figure 1c*), in line with previous results (*Wutz et al., 2020*). Upon RAD21 degradation, we again observed a stronger effect on chromatin-bound SA2 levels compared to SA1, suggesting that SA2 is more sensitive to cohesin loss than SA1. Residual SA proteins retained their ability to interact with CTCF in the absence of RAD21 and additionally, the interaction between SA1 and CTCF was further enhanced (*Figure 1c*). Reciprocal coIPs with CTCF confirmed the CTCF-SA interactions in RAD21-depleted cells, including the differences between SA1 and SA2 (*Figure 1d*). We validated these results in a second cell line and using siRNA-mediated knockdown (KD) of SMC3 (*Figure 1—figure supplement 1c*). We also confirmed that approximately 20% of SA1 and SA2 were bound to chromatin without RAD21 even in unperturbed RAD21^mAC HCT116 cells (*Figure 1—figure supplement 1d*), reminiscent of reports of that SA1 can be independent of cohesin at telomeres (*Bisht et al., 2013*).

Next, we performed two-color Stochastic Optical Reconstruction Microscopy (STORM) to assess the nuclear distribution and co-localization of SA1, SA2 and CTCF with nanometric resolution in control and RAD21-degraded cells (*Figure 1e* and *Figure 1—figure supplement 1e*). In control RAD21^mAC cells, we observed clustering of CTCF, SA1 and SA2 localizations as quantified by cluster analysis and nearest neighbor distance (NND) analysis of protein clusters (*Ricci et al., 2015*). SA1 and CTCF exhibited higher densities compared to SA2, with shorter distances between clusters (mean NND of 68.9, 65.3, and 78.9 nm for CTCF, SA1 and SA2, respectively) (*Figure 1f and g*). Furthermore, we analyzed the relative distribution of SA clusters to CTCF clusters by assessing the NND distribution between SA1 and CTCF, and SA2 and CTCF. Both SA1 and SA2 exhibited significant co-localization with CTCF at short distances in RAD21^mAC cells (*Figure 1h* and *Figure 1—figure supplement 1f, g*).

Upon IAA treatment, we observed a decreased density of detected SA1, SA2, and CTCF in two analyzed clones (*Figure 1e and f* and *Figure 1—figure supplement 1e*), suggesting that RAD21 degradation affects the stability of SA proteins and CTCF. As observed by conventional confocal microscopy (*Figure 1b*), SA2 localizations were more affected than SA1 (mean density reduction in SA1, 32% and SA2, 42% compared to EtOH controls). Accordingly, SA1, SA2, and CTCF clusters were more sparsely distributed across the nucleus upon RAD21 degradation (mean NND 81.1, 109.7 and 117.5 nm, respectively) with SA2 significantly more affected than SA1 (mean NND increase 24.3%, 39% respectively) (*Figure 1g*). SA proteins and CTCF remained co-localized upon RAD21 degradation as compared to both the control cells and to a simulation of randomly-distributed protein clusters at the same density (*Figure 1h* and *Figure 1—figure supplement 1f, g*). Interestingly, while the

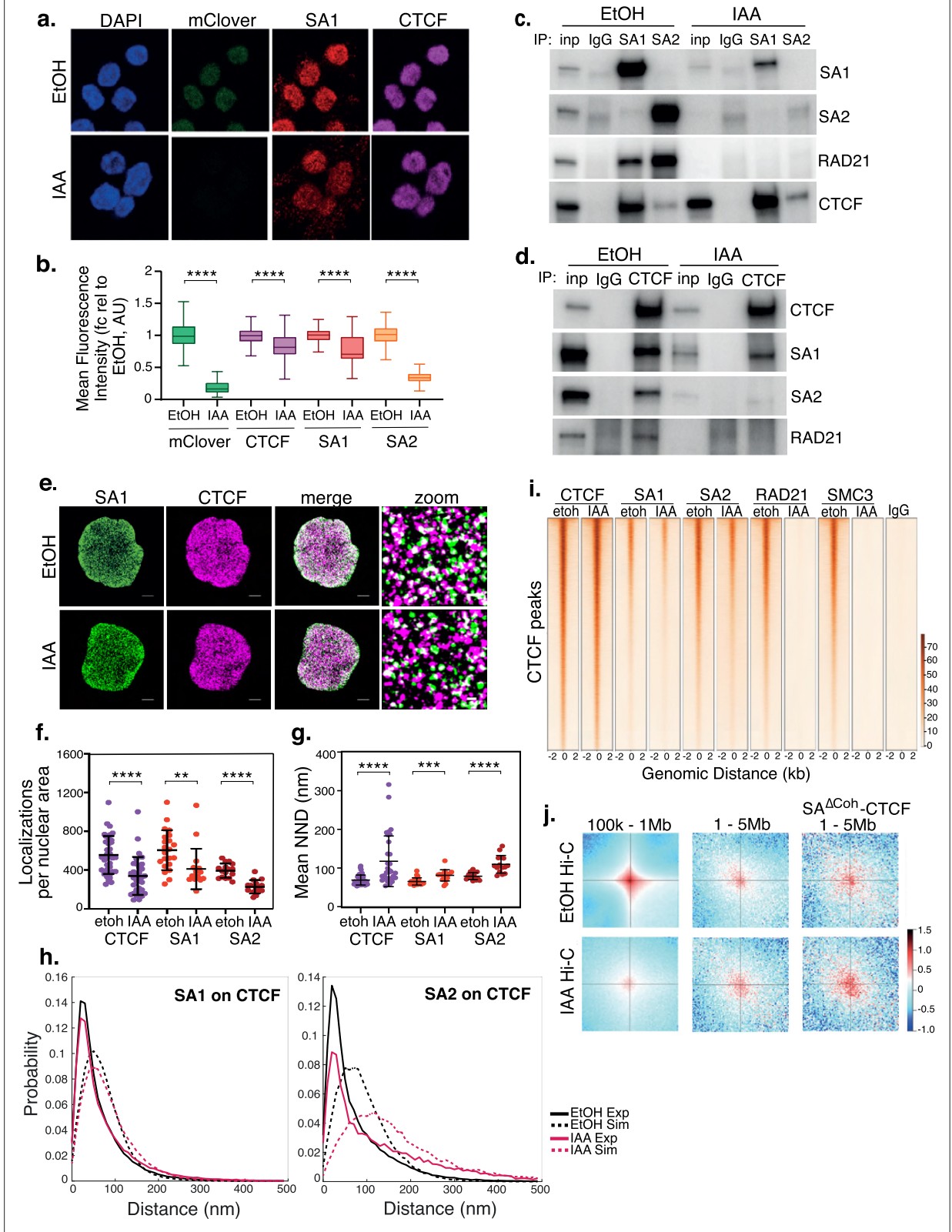

**Figure 1.** SA interacts with CTCF in the absence of cohesin. (**a**) Representative confocal images of SA1 and CTCF IF in RAD21$^{mAC}$ cells treated with ethanol (EtOH) as a control or Auxin (IAA) for 4 hr. Nuclei were counterstained with DAPI. (**b**) Imaris quantification of the relative mean fluorescence intensity (MFI) of mClover, CTCF, SA1 and SA2 in EtOH and IAA-treated RAD21$^{mAC}$ cells. Whiskers and boxes indicate all and 50% of values, respectively. Central line represents the median. Asterisks indicate a statistically significant difference as assessed using two-tailed t-test. **** p<0.0001. n>50 cells/

*Figure 1 continued on next page*

*Figure 1 continued*

condition from three biological replicates. Chromatin coIP of (**c**) SA1, SA2, and IgG with RAD21 and CTCF or (**d**) CTCF and IgG with RAD21, SA1, and SA2 in RAD21$^{mAC}$ cells treated with EtOH or IAA for 4 hr. Input represents (**c**) 2.5% and (**d**) 1.25% of the material used for immunoprecipitation. (**e**) Dual-color STORM images of SA1 (green) and CTCF (magenta) in EtOH and IAA-treated RAD21$^{mAC}$ cells. Representative full nuclei and zoomed nuclear areas are shown. Line denotes 2 microns and 200 nm for full nuclei and zoomed areas respectively. See figure supplements for SA2 STORM images. (**f**) Mean CTCF, SA1 and SA2 localization densities (localizations normalized with nuclear area) in EtOH and IAA-treated RAD21$^{mAC}$ cells (n = >30, >17, and>15 nuclei for CTCF, SA1, and SA2, respectively). Mean and SD are plotted, Mann Whitney test. ** p<0.005, *** p<0.0005, **** p<0.0001. (**g**) Mean Nearest Neighbor Distance (NND) of CTCF, SA1, and SA2 clusters in nanometers in EtOH and IAA-treated cells (n = >38, >14. and>23 nuclei for CTCF, SA1, and SA2, respectively). Mean and SD are plotted, Mann Whitney test. **** p<0.0001. (**h**) NND distribution plot of the distance between CTCF and SA1 (left panel) or SA2 (right panel) clusters in EtOH and IAA-treated cells. Experimental data are shown as continuous lines, random simulated data are displayed as dotted lines. (**i**) ChIP–seq deepTools heat map of CTCF, SA1, SA2, Rad21 and SMC3 binding profiles in control (EtOH) and IAA-treated RAD21$^{mAC}$ cells. Selected regions are bound by CTCF in control conditions. (**j**) Analysis of contact frequency hotspots from Hi-C libraries generated from EtOH-treated (top row) and IAA-treated (bottom row) RAD21$^{mAC}$ cells. Contact frequencies were calculated in two distance ranges of 100 kb – 1 Mb and 1–5 Mb. The last column includes contact frequencies specifically at SA-CTCF$^{\Delta Coh}$ binding sites.

The online version of this article includes the following source data and figure supplement(s) for figure 1:

**Source data 1.** Original, unedited western blots corresponding to *Figure 1*.

**Figure supplement 1.** SA interacts with CTCF in the absence of cohesin.

**Figure supplement 1—source data 1.** Original, unedited western blots for *Figure 1—figure supplement 1*.

probability of SA1 at CTCF is only modestly affected, SA2 at CTCF is more affected in IAA-treated cells (*Figure 1h*), in line with our previous observations. Together, our results confirm the maintained interaction and spatial co-localization patterns of SA proteins with CTCF and reveal a difference in SA paralog stability in the absence of the core cohesin trimer.

## Cohesin-independent SA proteins are localized at clustered regions in 3D

Previous analyses of the contribution of SA proteins to genome organization (*Kojic et al., 2018*; *Wutz et al., 2017*) were performed in cells containing cohesin rings, possibly obscuring a functional role for SA proteins themselves in genome organization. To determine if cohesin-independent SA proteins may function at unique locations in the genome, we investigated whether the residual SA-CTCF complexes (herein, SA-CTCF$^{\Delta Coh)}$ in IAA-treated RAD21$^{mAC}$ cells occupied the same chromatin locations as in control cells. Using chromatin immunoprecipitation followed by sequencing (ChIP-seq), we determined the binding profiles of CTCF, SA1, SA2, RAD21 and SMC3 in RAD21$^{mAC}$ cells treated with EtOH or IAA. Pairwise comparisons of CTCF ChIP-seq with RAD21 or SA in control RAD21$^{mAC}$ cells revealed the expected overlap in binding sites (*Figure 1i* and *Figure 1—figure supplement 1h*). In contrast, both global and CTCF-overlapping RAD21 and SMC3 ChIP-seq signals were dramatically lost in IAA-treated cells (*Figure 1i* and *Figure 1—figure supplement 1h*). In agreement with our microscopy and biochemistry results, we detected residual SA1 and SA2 binding sites in IAA-treated cells which retained a substantial overlap with CTCF. Furthermore, the sites co-occupied by CTCF and SA proteins in RAD21-depleted cells were also bound in control conditions, were enriched at TSSs and were characterized by active chromatin marks including phospho-S5 POLR2A, H3K4me3, and H3K27ac (*Figure 1i* and *Figure 1—figure supplement 1i*). Thus, CTCF and SA maintain occupancy at their canonical binding sites in the absence of RAD21, suggesting that SA interaction with CTCF in the absence of the cohesin ring is a step in normal cohesin activity.

While depletion of cohesin results in a dramatic loss of Topologically Associated Domain (TAD) structure (*Rao et al., 2017*), the frequency of long-range inter-TAD, intra-compartment contacts (LRC) is increased (*Sofueva et al., 2013*; *Rao et al., 2017*), and enriched for CTCF (*Sofueva et al., 2013*) or active elements (*Rao et al., 2017*). To determine whether residual, chromatin-bound SA could be associated with LRCs in the absence of RAD21, we re-analysed Hi-C data from control and IAA-treated RAD21$^{mAC}$ cells (*Rao et al., 2017*). We quantified all contacts within two different scales of genome organization; local TAD topology (100 k-1Mb) and clustered LRCs (1–5 Mb) (*Figure 1j*). As previously shown (*Rao et al., 2017*), local TAD contacts are lost and clustered LRCs are enriched in IAA conditions. We probed the Hi-C datasets for contacts containing the residual SA-CTCF$^{\Delta coh}$ binding sites and observed a further enrichment in IAA conditions (*Figure 1j*, bottom right), indicating that SA-CTCF$^{\Delta coh}$ are enriched at the clustered LRCs formed when cells are depleted of cohesin and thus implicating

them in 3D structural configurations. Our results suggest that cohesin-independent SA, either with CTCF or alone, may itself contribute to higher order arrangement of active chromatin and regulatory features in 3D space.

## SA interacts with diverse 'CES-binding proteins' in cohesin-depleted cells

SA proteins contain a highly conserved domain known as the 'stromalin conservative domain' (SCD) (*Orgil et al., 2015*; *Roig et al., 2014*), or the 'conserved essential surface' (CES). Structural analysis of CTCF-SA2-RAD21 has shown that a F/YXF motif in the N-terminus of CTCF engages with a composite binding surface containing amino acids from RAD21 and the CES of SA2, forming a tripartite inter-action patch (*Li et al., 2018*). Further, the authors identified a similar F/YXF motif in other cohesin regulatory proteins, predicting interactions between the SA-RAD21 binding surface and additional chromatin proteins. To investigate whether SA proteins could associate with F/YXF-motif containing proteins beyond CTCF in the absence of RAD21 in vivo, we performed chromatin coIP with SA1 and SA2 in EtOH and IAA and probed for interaction with CTCF as before, and three additional F/YXF-motif containing proteins, CHD6, MCM3 and HNRNPUL2 (*Figure 2a* and *Figure 2—figure supplement 1a, b*). All the proteins interacted with SA1 in RAD21-control cells and their interaction with SA1 was enriched upon RAD21-degradation, indicating that the association was not dependent on the amino acids contributed by RAD21. Interestingly, despite SA2 also containing the conserved CES domain, the F/YXF-motif proteins showed little interaction with SA2 (*Figure 2a* and *Figure 2—figure supplement 1b*), pointing to the presence of additional features in SA1 that stabilize its interaction with F/YXF containing proteins in vivo. Together, our results confirm that SA, in particular SA1, can interact with proteins beyond just CTCF and reveal that RAD21 is not required for the interaction between SA and F/YXF proteins in vivo. These results prompted us to re-evaluate the role of SA in cohesin activity and consider possible novel functions for these proteins.

## SA1 interacts with a diverse group of proteins in the absence of cohesin

To delineate novel protein binding partners and putative biological functions of cohesin-independent SA1, we optimized our chromatin-bound, endogenous SA1 co-IP protocol to be compatible with mass-spectrometry (IP-MS) and used this to comprehensively characterize the SA1 protein–protein interaction (PPI) network in control and RAD21-degraded RAD21$^{mAC}$ cells. Three biological replicates were prepared from RAD21$^{mAC}$ cells that were either untreated (UT) or treated with IAA and processed for IP with both SA1 and IgG antibodies. In parallel, RAD21$^{mAC}$ cells were also treated with scrambled siRNAs or with siRNA to SA1 to confirm the specificity of putative interactors. Immunoprecipitated proteins were identified by liquid chromatography tandem mass spectrometry (LC-MS/MS). SA1 peptides were robustly detected in all UT and siCON samples and never detected in IgG controls, validating the specificity of the antibody. We used a pairwise analysis of IAA vs UT SA1 samples to generate a fold-change value for each putative interactor. These candidates were changed by at least 1.5-fold compared to UT controls, and sensitive to siSA1, yielding 136 high-confidence interactors whose abundance was significantly altered with RAD21 loss (SA1$^{\Delta Coh}$; *Figure 2b* and *Figure 2—figure supplement 1*). As expected, core cohesin subunits SMC1A and SMC3 were strongly depleted while no peptides were detected for RAD21. SA1 itself was significantly depleted compared to control cells, as were other cohesin regulators, known to directly interact with SA1, such as PDS5B (*Hons et al., 2016*). In line with the enrichment we observed for the CES-binding proteins in IAA-conditions (*Figures 1c and 2a*), the vast majority of the SA1$^{\Delta Coh}$ interactors were enriched for binding with SA1 in IAA conditions (117 of 136), and represent cohesin-independent SA1$^{\Delta Coh}$ interactors (*Figure 2b*).

We used STRING to analyse associations in the SA1$^{\Delta Coh}$ interactome and to identify enriched biological processes and molecular functions. We calculated enrichment relative to either the whole genome or to the SA1 interactome, which was determined in parallel from control cells (*Figure 2—figure supplement 1c*), revealing similar enrichment terms to previously published cohesin inter-actomes (*Kim et al., 2019*; *Panigrahi et al., 2012*), and validating our approach. Compared to the whole genome background, processes enriched in the presence of cohesin were still enriched in the SA1$^{\Delta Coh}$ PPI network and include a variety of functionally diverse cellular processes such as chromosome organization, transcription, RNA processing, ribosome biogenesis, and translation (*Figure 2c*

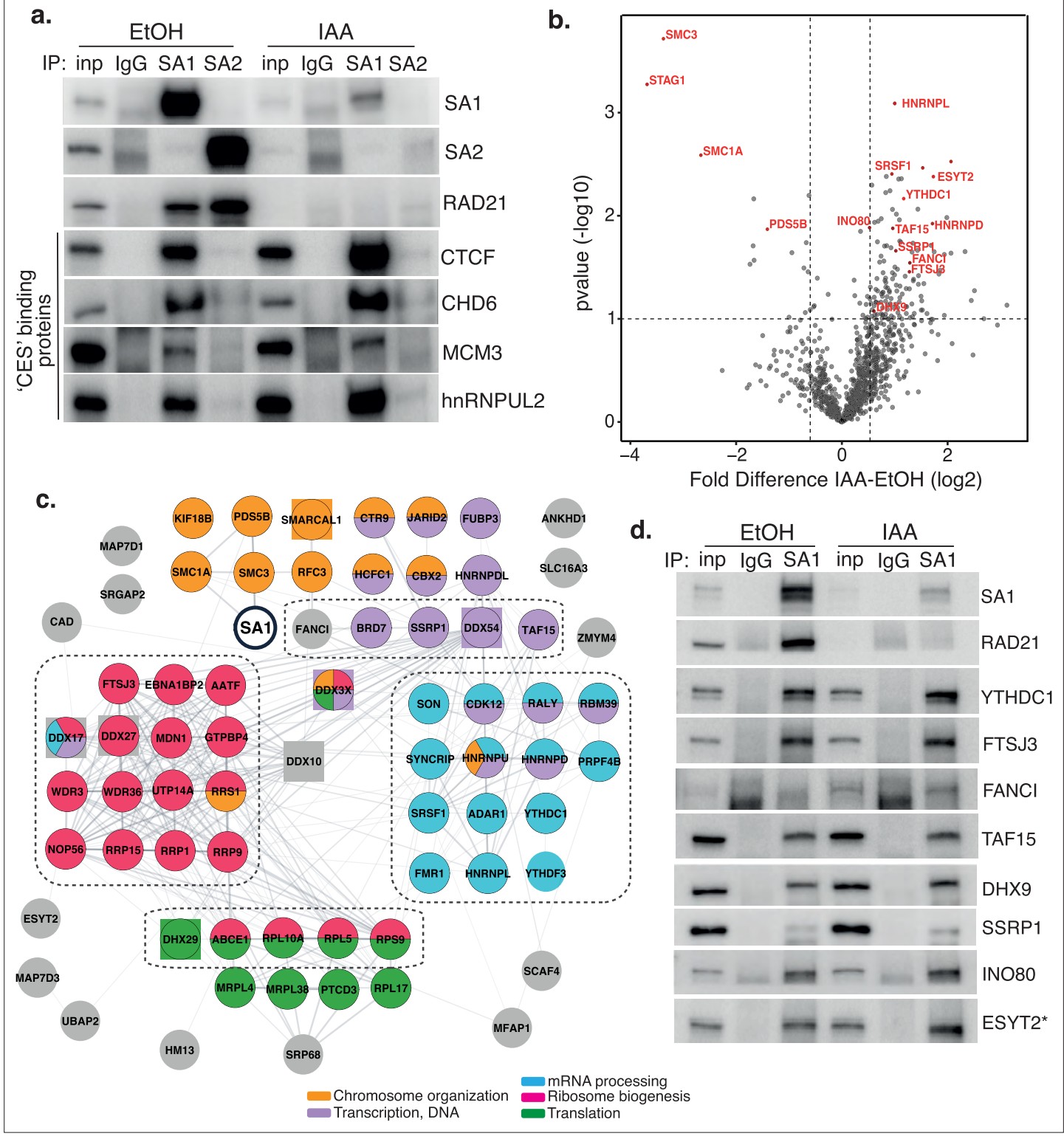

**Figure 2.** Characterization of SA1 protein-protein interaction network in RAD21-depleted cells. (**a**) Chromatin coIP of SA1, SA2, and IgG with four predicted CES-binding proteins in RAD21^mAC cells treated with EtOH or IAA for 4 hr. Input represents 1.25% of the material used for immunoprecipitation. (**b**) Volcano plot displaying the statistical significance (-log10 p-value) versus magnitude of change (log2 fold change) from SA1 IP-MS data produced from untreated or IAA-treated RAD21^mAC cells (n=3). Vertical dashed lines represent changes of 1.5-fold. Horizontal dashed line represents a pvalue of 0.1. Cohesin complex members and validated high-confidence proteins have been highlighted. (**c**) SA1^ΔCoh interaction network of protein–protein interactions identified in RAD21^mAC cells using STRING. Node colors describe the major enriched categories, with squares denoting

*Figure 2 continued on next page*

*Figure 2 continued*

helicases. Proteins within each enrichment category were subset based on p-value change in B. See figure supplements for full network. Dashed boxes indicate the proteins and categories which were specifically enriched in IAA-treatment compared to the SA1 interactome. (**d**) Chromatin IP of SA1 and IgG in RAD21^mAC cells treated with EtOH or IAA and immunoblotted with antibodies to validate the proteins identified by IP-MS. Input represents 1.25% of the material used for immunoprecipitation. * We note that ESYT2 is a F/YXF-motif containing protein.

The online version of this article includes the following source data and figure supplement(s) for figure 2:

**Source data 1.** Original, unedited Western blots for *Figure 2*.

**Source data 2.** MS stats values used in *Figure 2*.

**Figure supplement 1.** Characterization of SA1 protein-protein interaction network in RAD21-depleted cells.

**Figure supplement 1—source data 1.** Original, unedited western blots for *Figure 2—figure supplement 1*.

**Figure supplement 1—source data 2.** Sequence motifs for proteins tested in *Figure 2a*.

*and d*). Within this group, there are chromatin remodeling proteins (INO80 and SMARCAL1) and several transcriptional and epigenetic regulators such as JARID2 and TAF15. Similar to our ChIP and coIP results, this suggests that SA1 maintains interaction with proteins that localize with it in the presence of cohesin, albeit at different abundances.

RNA processing was the most enriched category in the SA1^{ΔCoh} PPI network (FDR = $3.62 \times 10^{-39}$) and included proteins involved in RNA modification (YTHDC1, ADAR1, FTSJ3), mRNA stabilization and export (SYNCRIP, FMR1), and RNA splicing regulators (SRSF1, SON) (*Figure 2c*). We also found a significant enrichment for DNA and RNA helicases (FDR = $3.54 \times 10^{-08}$) as well as RNA binding proteins (FDR = $9.11 \times 10^{-11}$) within which were many HNRNP family members (HNRNPU, aka SAF-A). We also found a highly significant enrichment of proteins associated with ribosome biogenesis (FDR = $2.20 \times 10^{-30}$) including both large and small subunit components; rRNA processing factors and components of the snoRNA pathway (FDR = $4.39 \times 10^{-05}$). Finally, translation was significantly enriched as a biological process (p=$1.64 \times 10^{-06}$), with several cytoplasmic translation regulators identified as SA1^{ΔCoh} interactors (DHX29, GCN1L1). Among these is ESYT2 which is primarily found in the cytoplasm and contains a F/YXF-motif (*Figure 2c* and *Figure 2—figure supplement 1d, e*). We validated eight of the highest-ranking proteins within the enriched functional categories described above in EtOH and IAA-treated RAD21^mAC cells (*Figure 2d*). Importantly, the enrichment of these proteins with SA1 in the IAA condition suggests that SA may have a role in these processes independently of the core cohesin complex.

Comparison of the SA1^{ΔCoh} interactome with the SA1 interactome revealed that the proteins involved in RNA processing (FDR = 0.0298), ribosome biogenesis (0.0197), ribonucleoprotein complex biogenesis (0.0298) and rRNA processing (0.0409) were enriched with SA1 following IAA treatment compared to SA1 in the presence of RAD21 (*Figure 2c*, dotted lines). Overall, our results show that SA1^{ΔCoh} PPIs contain not only transcriptional and epigenetic regulators, but are predominantly enriched for proteins with roles in RNA processing and modification, ribosome biogenesis and translation pathways. Thus, SA1 is involved in several biological processes and may facilitate an aspect of cohesin regulation at a variety of functionally distinct locations.

## SA proteins bind RNA independently of cohesin

Since RNA binding and RNA processing were among the most enriched categories in the SA1^{ΔCoh} PPI network, we hypothesized that SA proteins may also bind RNA. We performed SA-crosslinking and immunoprecipitation (CLIP) in untreated RAD21^mAC cells and found that both SA1 and SA2 directly bound RNA (*Figure 3a and b*). This was evidenced by detection of RNPs of the expected molecular weights, with a smear of trimmed RNA, which was stronger in the +UV and+PNK conditions, increased as the RNaseI concentration was reduced, and which was lost after siRNA-mediated SA KD (*Figure 3—figure supplement 1a–c*). We repeated the experiment in EtOH- and IAA-treated RAD21^mAC cells to determine if the SA subunits can bind RNA in the absence of cohesin. As before, RAD21 depletion reduced SA1 and SA2 (*Figure 3—figure supplement 1d*) and the amount of RNA crosslinked remained proportional to the amount of residual SA1 and SA2 protein (*Figure 3c* and *Figure 3—figure supplement 1e*), demonstrating that cohesin is not required for the interaction of these proteins with RNA in cells. Thus, cohesin-independent SA proteins interact with a wide array of RNA binding proteins (RBPs) as well as with RNA itself.

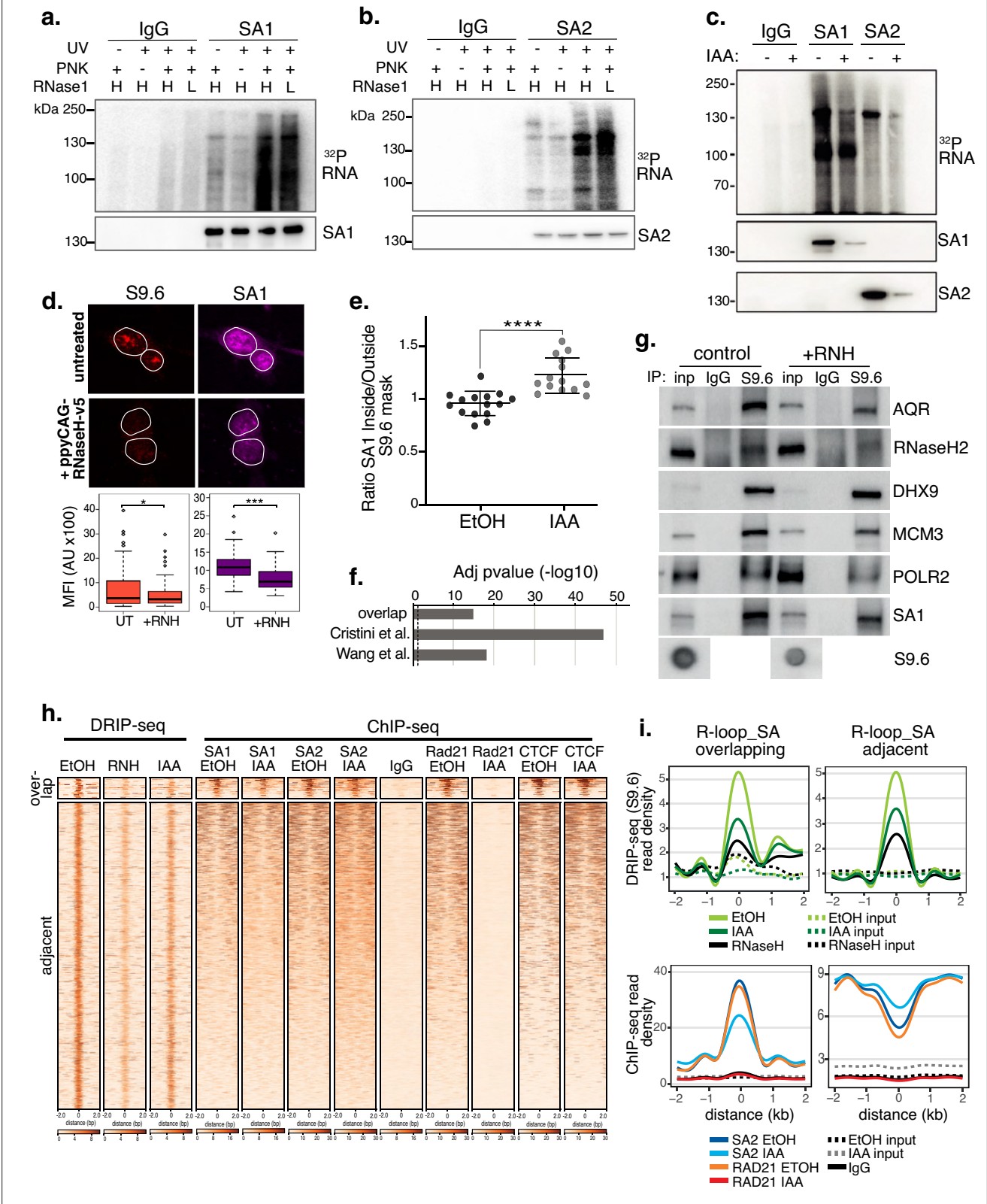

**Figure 3.** SA proteins bind to RNA and localize to R-loops. CLIP for (**a**) SA1, (**b**) SA2 and IgG controls. Autoradiograms of crosslinked $^{32}$P-labeled RNA are shown at the top and the corresponding immunoblots, below. CLIP was performed with and without UV crosslinking and polynucleotide kinase (PNK) and with high (H; 1/50 dilution) or low (L; 1/500 dilution) concentrations of RNase I. (**c**) CLIP for (a) SA1, SA2 and IgG control in EtOH (-) or IAA-treated (+) Rad21$^{mAC}$ cells. $^{32}$P-labeled RNA and the corresponding immunoblots are shown as above. (**d**) Top, Representative confocal images of S9.6

*Figure 3 continued on next page*

*Figure 3 continued*

and SA1 IF in RAD21^mAC cells untreated or overexpressing ppyCAG-RNaseH-v5. Expressing cells were identified with v5 staining. Nuclear outlines (white) are derived from DAPI counterstain. Bottom, Imaris quantification of the relative mean fluorescence Intensity (MFI) of S9.6 and SA1. Data are from two biological replicates with >75 cells counted/condition. Quantifications and statistical analysis were done as above. (**e**) STORM analysis of localization density for SA1 in S9.6 masks in EtOH and IAA. Ratio of SA1 localizations inside and outside S9.6 masks is shown. Ratio of above 1 represents an enrichment within the S9.6 domain. Mean and SD are plotted, statistics based on One-Way Anova test. Data are from two biological replicates. (**f**) -log10 transformed adjusted p-value (FDR) for enrichment of S9.6 interactome data from Cristini et al. and Wang et al., with the SA1^ΔCoh interactome. Overlap indicates the proteins identified in both of the S9.6 interactome datasets, representing a high confidence R-loop interactome list. (**g**) Chromatin coIP of S9.6 and IgG in RAD21^mAC cells treated with RNase H and immunoblotted with antibodies representing known R-loop proteins, as well as SA1. Input represents 1.25% of the material used for immunoprecipitation. Bottom, S9.6 dot blot of lysates used in coIP. (**h**) deepTools heatmap of DRIP-seq and ChIP-seq from RAD21^mAC cells. DRIP-seq was carried out in control (EtOH), RNase H (RNH), and IAA-treated cells. ChIP-seq was carried out for SA1, SA2, RAD21, CTCF, and IgG in EtOH and IAA-treated cells. Regions were selected based on DRIP-seq sensitivity to RNH and proximity with SA1 ChIP-seq. BEDTools identified regions of overlap or adjacent SA1 co-binding. (**i**) Summary plots showing mean DRIP-seq (top) or ChIP-seq (bottom) read density across the regions from (**h**), including sites of R-loop and SA 'overlap' (Left) or 'adjacent' (right) regions. Input samples are indicated with dotted lines.

The online version of this article includes the following source data and figure supplement(s) for figure 3:

**Source data 1.** Original, unedited western blots for *Figure 3*.

**Figure supplement 1.** SA proteins bind to RNA and localize to R-loops.

**Figure supplement 1—source data 1.** Original, unedited western blots for *Figure 3—figure supplement 1*.

## SA proteins localize to endogenous R-loops in the absence of cohesin

Proteins involved in RNA processing, such as splicing, modification and export, act as regulators of R-loops (*Santos-Pereira and Aguilera, 2015*). Furthermore, R-loops accumulate at sites of multiple biological processes including transcription, DNA replication and DNA repair (*Santos-Pereira and Aguilera, 2015*). As many of these processes were enriched in the SA1 interactome, we reasoned that the diversity of biological processes represented in the SA1^ΔCoh PPI network may be reflective of a role for SA proteins in R-loop biology.

We performed a number of experiments to investigate the localization of SA proteins at endogenous R-loops. First, we found a correlation between global SA and R-loop levels. We depleted endogenous R-loops by overexpressing ppyCAG-RNaseH-V5 in HCT116 cells. IF using the R-loop specific antibody S9.6 revealed that nuclear S9.6 levels were significantly reduced in cells which expressed V5 (38% of controls, p=0.04) and that mean SA1 signal was significantly reduced by 29% in the same cells (*Figure 3d*). Furthermore, RAD21^mAC cells treated with scramble control siRNAs or Smartpool (SP) siRNAs to AQR (a known suppressor of R-loops *Sollier et al., 2014*), SA1 or SA2 revealed that S9.6 IF signal was significantly increased in siAQR and siSA1 but not siSA2 cells compared to the siScr control (mean S9.6 signal increased by 28%, p=0.0004; 32%, p=3.90E-8; reduced by 10%, p=0.17, respectively; *Figure 3—figure supplement 1f*). Although S9.6 signal was reduced by IF in RAD21^mAC cells treated with IAA, this did not represent a significant change using this method (*Figure 3—figure supplement 1g*).

We also performed STORM imaging on EtOH and IAA-treated RAD21^mAC cells to assess the nuclear distribution of SA1 in the context of R-loops with and without RAD21. We measured the ratio of the SA1 signal inside and outside of the S9.6 signal mask. A ratio of 1 indicates a random distribution of SA1 with respect to S9.6 domains while a ratio above 1 reflects enrichment within S9.6 domains. In EtOH conditions, we did not detect enrichment of SA1 localizations, in fact SA1 was modestly depleted (mean ratio 0.93). However, upon IAA treatment, we observed a significant enrichment of SA1 localizations within S9.6 domains (mean ratio 1.24, p<0.0001; *Figure 3e*), strongly suggesting that SA1 proteins are localized within R-loop domains independently of cohesin.

In addition, we returned to our IP-MS experiment to analyse enrichment of R-loop-associated proteins in our SA1^ΔCoh interactome. We overlapped the proteins identified in two independent IP-MS experiments for R-loop interactors (*Cristini et al., 2018*; *Wang et al., 2018*) to create a high-confidence 'R-loop interactome' and then used a hypergeometric distribution to determine the significance of this category in the SA1^ΔCoh interactome (Methods). Both the custom R-loop interactome (termed 'overlap') as well as proteins from the individual studies were highly over-enriched in the SA1^ΔCoh interactome (FDR = $1.1 \times 10^{-15}$, $1.4 \times 10^{-47}$, $7.7 \times 10^{-19}$, respectively; *Figure 3f*). To directly measure this, we optimized a coIP method using the S9.6 antibody in RAD21^mAC cells (*Figure 3g* and *Figure 3—figure supplement 1h*). In agreement with published results, we found that S9.6 precipitated the known

R-loop helicases AQR, DHX9, RNase H2 (*Sollier et al., 2014*; *Cristini et al., 2018*) as well as MCM3 and RNA Pol II (POLR2) (*Skourti-Stathaki et al., 2014*). Both SA1 and SA2 precipitated with S9.6 and treatment with RNase H (RNH) revealed the specificity of the S9.6-SA interactions since the reduction of R-loop signal was proportional to the observed reduction in coIP of SA1 by S9.6 (*Figure 3g* and *Figure 3—figure supplement 1h, i*).

Finally, we used a high-resolution, genome-wide method to detect R-loops in HCT116 cells. RAD21^mAC cells were treated with RNH to confirm the specificity of our method and with EtOH or IAA to assess the impact of cohesin loss on R-loops and subjected to DNA-RNA Immunoprecipitation coupled with sequencing (DRIP-seq) using the S9.6 antibody. We combined these datasets with our ChIP-seq for SA proteins, RAD21 and CTCF in EtOH or IAA conditions to confirm the associations described above. We detected 50,338 RNH-sensitive R-loop sites which were also sensitive to acute degradation of RAD21, albeit not to the same extent as RNH treatment (average S9.6 signal was reduced by 31.4% in RNH and 16.8% in IAA compared to EtOH control; *Figure 3h and i*). Among the RNH-sensitive R-loop sites, we detected two regimes of SA-R-loop biology. A small proportion of R-loop sites directly overlapped with SA1/2, RAD21 and CTCF in control EtOH conditions. These sites were enriched at genes and both the SA1 and SA2 read density was sensitive to RAD21 loss (*Figure 3h and i* and *Figure 3—figure supplement 1j*, k). On the other hand, a larger proportion of R-loops had SA signals adjacent (bound within 2 kb of the R-loop peak). Interestingly, these SA sites were enriched in repressed chromatin and were not sensitive to RAD21 loss, in fact their read density was enriched compared to EtOH controls (*Figure 3h and i* and *Figure 3—figure supplement 1j, k*), reminiscent of the enrichment observed previously by STORM imaging (*Figure 3e*).

## NIPBL-independent cohesin loading mediated by SA proteins

Our results thus far revealed that SA^ΔCoh is localized to clustered regions, engages with RNA and various RBPs and is localized to R-loops. Several lines of evidence suggest that alongside the canonical NIPBL/Mau2 loading complex, SA proteins contribute to cohesin's association with chromatin (*Murayama and Uhlmann, 2014*; *Orgil et al., 2015*) and that its functions may go beyond simply acting as a bridging protein (*Orgil et al., 2015*). Thus, we hypothesized that SA proteins support genome organization in their own right and therein facilitate cohesin's association with chromatin.

The RAD21^mAC system has the advantage that when IAA is washed-off cells, the RAD21 protein is no longer degraded and can become 'reloaded' onto chromatin. We assessed this by measuring mClover signal intensity using IF and observed that it was robustly lost in IAA conditions and was partially restored to EtOH levels within 4 hr of IAA withdrawal (*Figure 4a and b* and *Figure 4—figure supplement 1a*). We note the spatial distribution of RAD21 was itself variable, ranging between highly compartmentalized and randomly distributed (*Figure 4a and c*). This provided a unique opportunity to assess how SA influences cohesin reloading in vivo and the potential role for RNA and R-loops in this process.

We assessed reloading using both single-cell and bulk methods, coupled with siRNA-mediated KD to determine how specific proteins affected cohesin reloading in vivo. We first measured the impact of the canonical cohesin loader, NIPBL. RAD21^mAC cells were treated with scramble or NIPBL siRNAs and subsequently grown in EtOH or IAA. The '0 h' and '4 h' post EtOH/IAA wash-off samples represent the extent of cohesin *degradation* or *reloading*, respectively (*Figure 4—figure supplement 1b*). Chromatin fractionation in high-salt conditions followed by immunoblot analysis confirmed the loss of the loader complex, NIPBL and MAU2 [known to become destablized upon NIPBL loss (*Watrin et al., 2006*)]. As expected, in NIPBL KD conditions, mean RAD21 re-loading efficiency was reduced, although surprisingly, this was incomplete (58% of siCon; mean re-loading siNIPBL, 2.1 vs siCon, 3.6), and did not represent a statistically significant difference (p=0.33) (*Figure 4d and e* and *Figure 4—figure supplement 1c*). This result was reproduced using IF, where mean mClover signal in siNIPBL-treated cells was 54.9% of siCon (MFI siCon, 6563 vs siNIPBL, 3602) (*Figure 4g*), indicating that cells can still load cohesin in the absence of NIPBL.

We reasoned that SA proteins may be contributing to the observed NIPBL-independent reloading. Thus, we repeated the experiments to include siRNA to SA1 and SA2 together (siSA), and a siNIPBL +siSA condition. In both population and single cell analysis of reloading, SA KD had a more dramatic effect on cohesin re-loading efficiency than NIPBL KD, reducing RAD21 on chromatin to 37% of siCon (mean siSA, 1.9 vs siCon, 5.1, p=0.002 for (*Figure 4f* and *Figure 4—figure supplement*

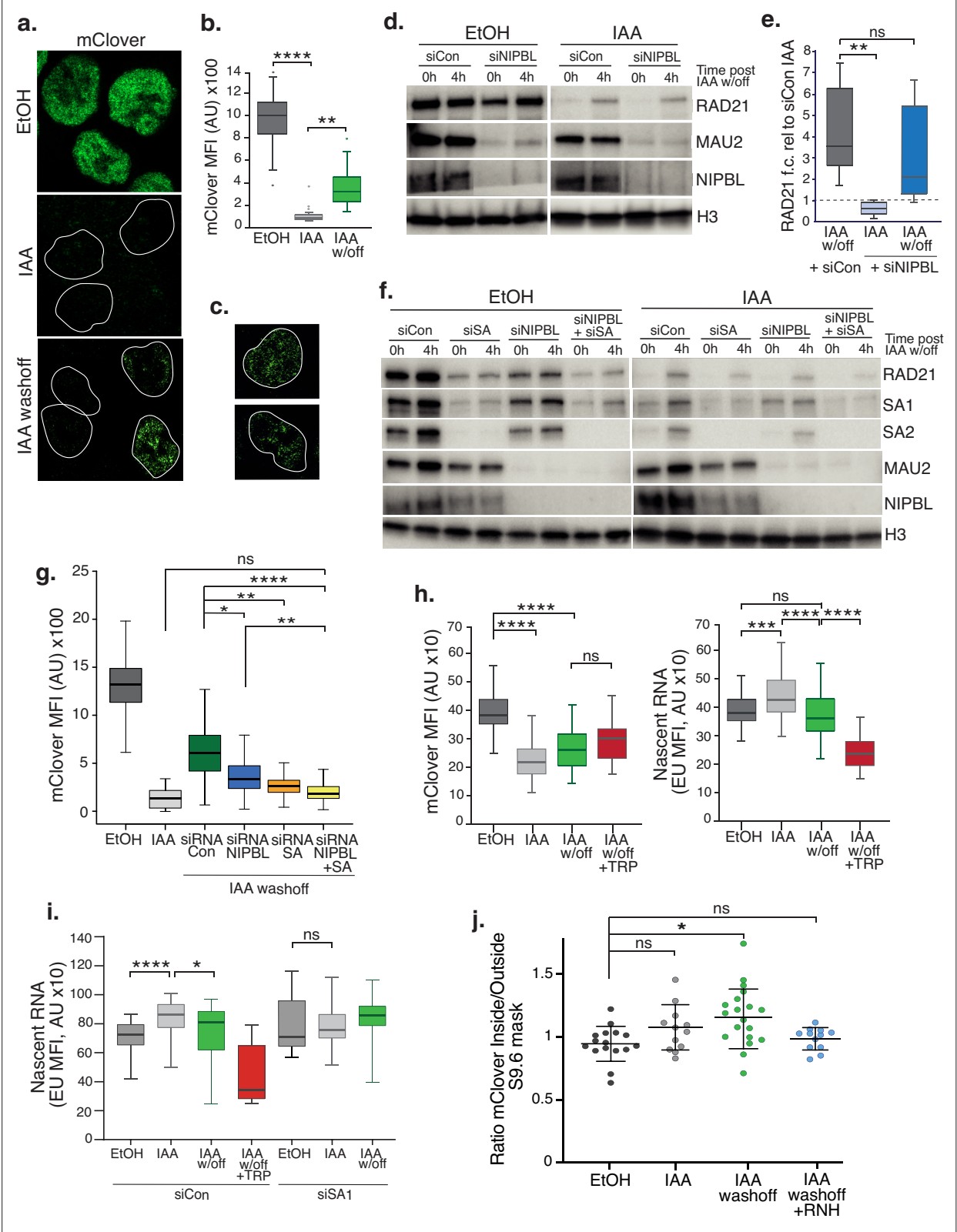

**Figure 4.** SA proteins contribute to cohesin loading. (**a**) Representative confocal images of immunofluorescence for mClover in EtOH, IAA and IAA washoff conditions. White lines denote nuclei based on DAPI staining. (**b**) Imaris quantification of the mean fluorescence intensity (MFI) of mClover in EtOH, IAA-treated and IAA washoff RAD21^mAC cells. Analysis and statistics as before. n>50 cells/condition from two biological replicates. (**c**) Examples of individual cells 4 hr post IAA washoff showing different distributions of mClover signal within the nucleus. White lines denote nuclei based on

*Figure 4 continued on next page*

*Figure 4 continued*

DAPI staining. (**d**) Representative immunoblot analysis of chromatin-bound RAD21, MAU2 and NIPBL levels in RAD21^mAC cells treated with scramble control siRNA (si Con) or siRNA to NIPBL followed by EtOH or IAA treatment. 0 h and 4 h represent no wash-off of IAA or a sample taken 4 hr after washoff of IAA (the 'reloading timepoint). H3 was used as a loading control. *NB* The full blots are in *Figure 4—figure supplement 1c*. (**e**) Western blot densitometry quantification. RAD21 fold change relative to siCon samples at the 0 h timepoint in siCon 4 h (grey), siNIPBL 0 hr (light blue), and siNIPBL 4 hr (dark blue). Whiskers and boxes indicate all and 50% of values, respectively. Central line represents the median. Statistical analysis as assessed using a two-tailed t-test. Data is from eight biological replicates. (**f**) Representative immunoblot analysis of chromatin-bound RAD21, SA1, SA2, MAU2, and NIPBL levels in RAD21^mAC cells treated according to the schematic described in *Figure 4—figure supplement 1b* and including samples treated with siRNA to SA1 and SA2 together (siSA) and siRNA to NIPBL +siSA. H3 was used as a loading control. Quantifications can be seen in in *Figure 4—figure supplement 1d*. (**g**) Imaris quantification of the mClover MFI in EtOH, IAA-treated and IAA washoff RAD21^mAC cells treated with siRNA to NIPBL, SA1/2 and siRNA to NIPBL +siSA. Asterisks indicate a statistically significant difference as assessed using two-tailed T-test. Data is from two biological replicates with >50 cells per experiment. (**h**) Imaris quantification of the mClover MFI (left) and RNA (based on EU incorporation (right) in EtOH and IAA-treated RAD21^mAC cells. Whiskers and boxes indicate all and 50% of values, respectively. Central line represents the median. Asterisks indicate a statistically significant difference as assessed using one-way ANOVA. n>50 cells/condition from two biological replicates.(**i**) Analysis of mClover MFI as in h) above, this time treated with siRNA to SA1/2 or scrambled controls. Analysis and statistics as above. n>50 cells/condition from two biological replicates. (**j**) STORM analysis of localization density for RAD21-mClover in S9.6 masks in EtOH (black), IAA (grey), and IAA washoff (green) conditions. Ratio of mClover localizations inside and outside S9.6 masks is shown. Ratio of above 1 represents an enrichment within the S9.6 domain. Mean and SD are plotted, statistics based on One-Way ANOVA test. Data are from two biological replicates.

The online version of this article includes the following source data and figure supplement(s) for figure 4:

**Source data 1.** Original, unedited western blots for *Figure 4*.

**Figure supplement 1.** SA proteins contribute to cohesin loading.

**Figure supplement 1—source data 1.** Original, unedited western blots for *Figure 4—figure supplement 1*.

---

*1d*). MFI siCon, 6563 vs siSA, 2303 p<0.0001 for *Figure 4g*). In the absence of both SA and NIPBL, cohesin reloading was reduced further (mean siNIPBL +siSA, 1.4 vs siCon, 5.1, p=0.001 for (*Figure 4f* and *Figure 4—figure supplement 1d*). MFI siCon, 6563 vs siNIPBL +SA,1925 p<0.001 for *Figure 4g*), indicating that SA performs an important and complementary step to NIPBL during normal reloading. Given the differences between SA1 and SA2 reported herein, we also performed the reloading experiment to separate the effects of SA1 and SA2. As expected from our co-IP results (*Figure 1c*), RAD21 levels in RAD21^mAC cells were more affected by siSA2 than siSA1 (*Figure 4—figure supplement 1e*). We observed that cohesin reloading was more efficient in siSA2 (where SA1 is present) than in siSA1 (where SA2 is present), and that siSA1 was similar in reloading to siSA (*Figure 4—figure supplement 1e*). Together these observations suggest that the bulk of the reloading in RAD21^mAC cells treated with IAA is supported by SA1.

## SA proteins stabilize nascent RNA in the absence of cohesin

Given the association of SA proteins with RNA and RBPs and the dependence of cohesin reloading on SA1, we tested the requirement for RNA in cohesin reloading. Cells were treated as above with a pulse of 5 ethynyl uridine (EU) prior to collection. EU becomes actively incorporated into nascent RNA and can be measured by IF alongside the change in RAD21-mClover. While a significant reduction in nascent RNA signal was detected upon treatment with Triptolide (TRP), mClover signal was not significantly changed compared to IAA washoff conditions, indicating that RNA is not a key determinant of cohesin reloading per se (*Figure 4h*, left panel). However, we did observe a significant increase in nascent RNA upon acute RAD21 degradation which returned to EtOH levels when cohesin became reloaded onto chromatin (*Figure 4h*, right panel). These results pointed to the stabilization of nascent RNA in the absence of cohesin. Given the association of SA1^ΔCoh with RNA and RBP, we repeated the experiment in siSA1 KD conditions. Again, we observed an increase in nascent RNA in IAA conditions compared to EtOH control, but this was no longer detected in cells treated with siRNA to SA1 (*Figure 4i*). Our results point to a role for SA1 in the stabilization of nascent RNA in the absence of cohesin and a possible competition between SA-RNA and SA-cohesin associations.

Our results thus far showed that SA proteins remain chromatin associated in the absence of cohesin (*Figure 1*), when they bind RBP (*Figure 2*) and RNA and are localized to R-loops (*Figure 3*). We also report that SA1 proteins contribute to cohesin's re-association with chromatin and that this involves nascent RNA (*Figure 4*). Thus, we reasoned that SA may facilitate cohesin reloading at R-loops. It was technically challenging to measure reloading upon over-expression of ppyCAG-RNaseH. As

an alternative, we used STORM imaging to assess the nuclear distribution of the reloaded cohesin in the context of R-loop clusters by comparing EtOH- and IAA-treated to IAA-washoff RAD21$^{mAC}$ cells (*Figure 4j*). As before, we measured the ratio of signal (this time RAD21-mClover) inside and outside of the S9.6 mask. Interestingly, in EtOH conditions, RAD21 localizations were depleted from the S9.6 domain (mean ratio 0.95; *Figure 4j*) similar to what we observed for SA1 (*Figure 3h*). Since STORM is such a sensitive approach, trace localizations of mClover will always be detected, even in IAA conditions when the bulk of the signal is lost. The few localizations we observed were indeed modestly enriched within the S9.6 mask, although these were not significantly different from EtOH (mean ratio 1.08, p=0.10). These localizations may represent either extremely stable or freshly loaded cohesin. Upon IAA washoff, new RAD21-mClover molecules are readily detected, became significantly enriched within S9.6 domains compared to EtOH treated cells (mean ratio 1.19, p=0.029) and were sensitive to treatment with RNase H (mean ratio 0.98; *Figure 4j*). Overall, our results point to a role for SA1 proteins in mediating reloading of cohesin at R-loops.

## A basic exon in the C-terminus of SA2 tunes interactions with RBPs

While both SA1 and SA2 played a role in cohesin's reloading, SA1 was the dominant paralog (*Figure 4—figure supplement 1e*). In addition, SA2 was not able to compensate for SA1 in R-loop stability (*Figure 3—figure supplement 1f*), despite its interaction with RNA (*Figure 3a and b*) and R-loops (*Figure 3—figure supplement 1h*). Previous publications have described association of RBP from SA2 MS-IP in HCT116 cells (*Kim et al., 2019*). Indeed, several of these RBPs overlap with the proteins described here as SA1 interactors (*Figure 2b and c*) and are enriched in SA1 IP in IAA conditions (*Figure 2a and d*). However, we did not observe robust enrichment of RBPs compared to input in SA2 IP, in either EtOH or IAA conditions. This was reminiscent of the differential interactions between SA1 and SA2 with F/YXF containing proteins (*Figure 2a*). These results thus raised the question of whether additional features in SA2 may be required to stabilize these interactions and functions.

*STAG1* and *STAG2* express transcript variants in RAD21$^{mAC}$ cells. We re-analysed publicly available RNA-seq datasets and quantified alternative splicing profiles using VAST-tools analysis (*Irimia et al., 2014*). We found that one prominent variant is highly conserved in human HCT116 cells AND mouse embryonic stem and neural progenitor cells, arises from the alternative splicing of a single C-terminal exon, exon 31 in SA1 (SA1$^{e31\Delta}$) and exon 32 in SA2 (SA2$^{e32\Delta}$; *Figure 5a*, inset). The significance of this splicing event is unknown. The majority of SA1 mRNAs *include* e31 (average 'percent spliced in' (PSI) 97.7% in HCT cells), while the majority of SA2 mRNAs *exclude* e32 (average PSI 20.4% in HCT; *Figure 5b*, *Table 1* and *Figure 5—figure supplement 1a, b*). We confirmed this at the protein level by designing custom esiRNAs to specifically target SA1 e31 or SA2 e32 (Methods). Smartpool (SP) KD reduced the levels of SA1 and SA2 to similar extents compared to scrambled controls (87% and 94%, respectively; *Figure 5c*). Specific targeting of SA1 e31 led to a reduction of 85% of SA1 compared to esiRNA control (which was comparable to SP KD). In contrast, SA2 e32 targeting had a minimal effect on SA2 protein levels compared to its esiRNA control (reduction of 2%; *Figure 5c*), in line with the PSI data (*Figure 5b*) and indicating that the dominant SA2 isoform does not contain e32.

**Table 1.** Published datasets used in this study.

| Accession no. | Analysis description | Publication DOI or Ref | Figure Reference |
|---|---|---|---|
| GSE104334 | Long-range contact analysis of Hi-C datasets | 10.1016 /j.cell.2017.09.026 | 1j |
| GSE89729 | Percent Spliced In (PSI) analysis of RNA-seq datasets | 10.1172/jci.insight.91419 | 5d, "HCT Zuo" |
| GSM958749 | Percent Spliced In (PSI) analysis of RNA-seq datasets | ENCODE HCT116 RNAseq | 5b, "HCT ENCODE" |
| GSM958735 | Percent Spliced In (PSI) analysis of RNA-seq datasets | ENCODE HeLa RNAseq | 5b, "HeLa" |

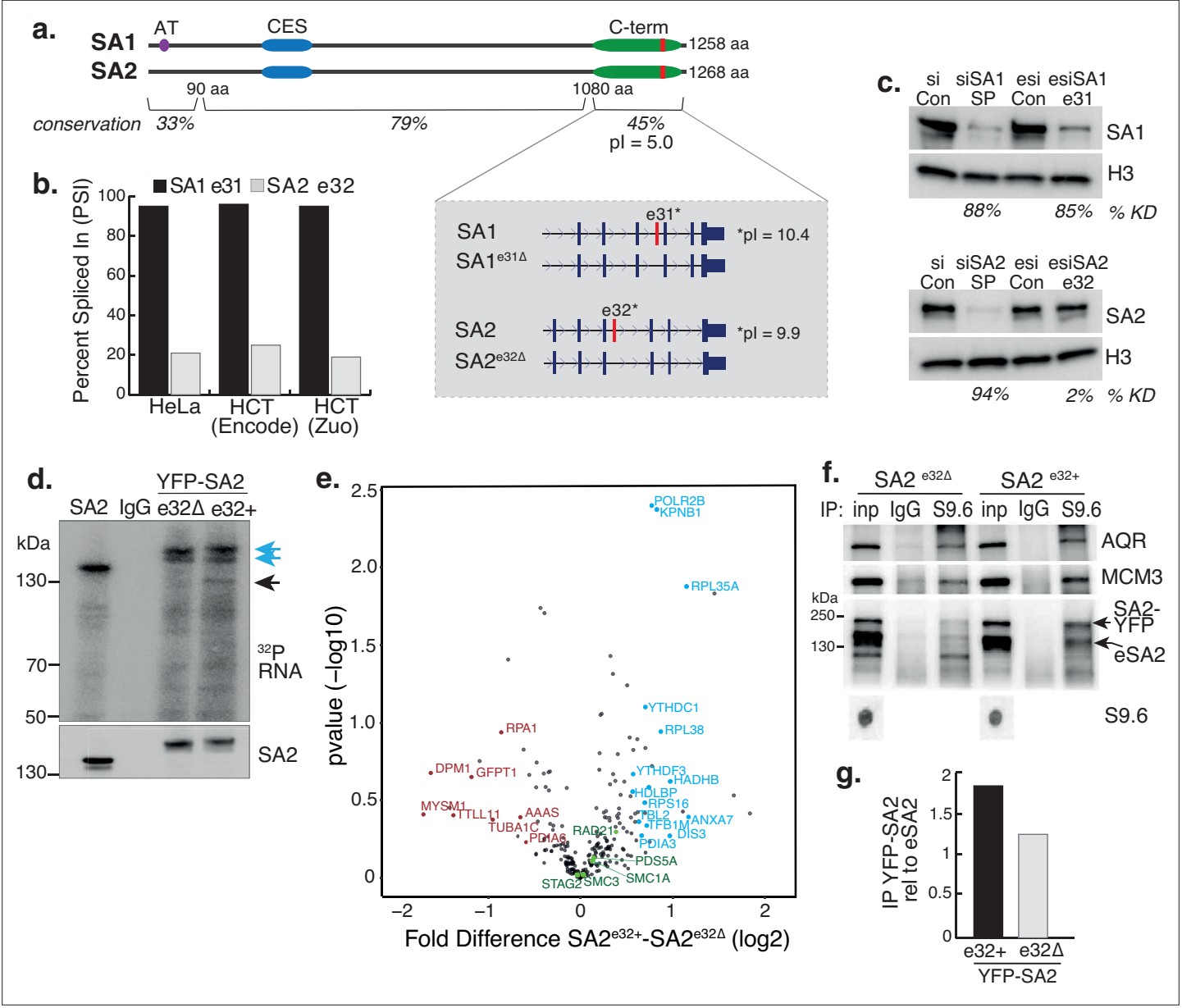

**Figure 5.** A basic exon in SA2 influences RBP stability. (**a**) Schematic of the SA1 and SA2 proteins showing the SA1-specific AT-hook, the conserved CES domain (blue) and the acidic C-terminus (green) which contains the basic alternatively spliced exon (red). Right-hand zoom-in indicates the spliced exons for SA1 (top) and SA2 (bottom) and the pI for each. The conservation scores for the divergent N- and C-termini and the middle portion of the proteins which contains the CES domain are shown. (**b**) Percent Spliced In (PSI) calculations for SA1 exon 31 (black) and SA2 exon 32 (grey) based on VAST-Tools analysis of RNA-seq from multiple datasets (see Methods). (**c**) (top) Immunoblot analysis of SA1 levels in chromatin lysates after treatment with scrambled siRNAs (siCon), SmartPool SA1 siRNAs (siSA1 SP), control esiRNAs (esiCon) and esiRNA designed to target SA1 exon 31 for 48 hr in RAD21[mAC] cells. (bottom) Immunoblot analysis of SA2 levels in chromatin lysates after treatment with scrambled siRNAs (siCon), SmartPool SA2 siRNAs (siSA2 SP), control esiRNAs (esiCon) and siRNA designed to target SA2 exon 32 for 48 hr in RAD21[mAC] cells. H3 serves as a loading control. The percentage of knockdown (KD) after SA signal is normalized to H3 is shown. (**d**) CLIP for endogenous SA2 and IgG control in cells in which either YFP-tagged SA2 containing exon 32 (e32+) or YFP-tagged SA2 lacking exon 32 were expressed for 48 hrs. CLIP reveals RNA associated with SA2 (blue arrows) and RBPs which specifically associate with exon-32 containing SA2 (black arrow). (**e**) Volcano plot displaying the statistical significance (-log10 p-value) versus magnitude of change (log2 fold change) from IP-MS of HCT116 cells expressing either YFP-SA2[e32+] or YFP-SA2[e32Δ] (n=3 biological replicate IP). Cohesin complex members are highlighted in green and the two most enriched functional categories of RNA-binding proteins in blue or Post-translational modification in red. (**f**) Chromatin coIP of S9.6 and IgG in RAD21[mAC] cells expressing the YFP-SA2 isoforms and immunoblotted with antibodies representing known R-loop proteins, as well as endogenous SA2 (eSA2). Input represents 1.25% of the material used for immunoprecipitation. Bottom, S9.6 dot blot of lysates used in coIP. *NB* the shift in SA2 signal representing overexpressed protein (SA2-YFP).

*Figure 5 continued on next page*

*Figure 5 continued*

(**g**) Quantification of the immunoblot signal from (**f**) of SA2 in the YFP-SA2 isoform band relative to input and to eSA2 signal. YFP-SA2$^{e32+}$ is more enriched by S9.6 IP compared to YFP-SA2$^{e32\Delta}$.

The online version of this article includes the following source data and figure supplement(s) for figure 5:

**Source data 1.** Original, unedited western blots for *Figure 5*.

**Source data 2.** MS stats values used in *Figure 5*.

**Figure supplement 1.** A basic exon in SA2 influences RBP stability.

**Figure supplement 1—source data 1.** Excel document of the PSI values for *Figure 5—figure supplement 1*.

**Figure supplement 1—source data 2.** Excel document of intensities of cohesin proteins for *Figure 5—figure supplement 1c*.

These results imply that cells 'tune' the availability of e31/32 in SA proteins, prompting us to investigate the nature of these exons. Interestingly, the amino acid (aa) sequence of the spliced SA exons encode a highly basic domain within an otherwise acidic C-terminus (*Figure 5a*, inset). Overall, the SA paralogs are highly homologous, however the N- and C-termini diverge in their aa sequence. Despite this, e31 and e32 have retained their basic properties (pI = 10.4 and 9.9, respectively). Basic patches can act as regulatory domains and bind nucleic acids prompting us to ask whether these alternatively spliced basic exons contribute to the association of SA proteins with RNA (*Figure 3a*). We cloned cDNAs from HCT116 cells representing the exon32-containing SA2 (SA2$^{e32+}$) and the canonical exon32-lacking SA2 (SA2$^{e32\Delta}$), tagged them with YFP, expressed them in HCT116 cells and purified the tagged isoforms to compare their ability to interact with RNAusing CLIP (*Figure 5d*). While the presence of e32 did not change the ability of SA2 to interact with RNA (*Figure 5d*, blue arrows), cells expressing the alternative exon routinely enriched RBPs with molecular weights ~110–140 kDa (*Figure 5d*, black arrow), strongly suggesting that the e32 domain may act to stabilize the association of SA2 with RBPs.

To identify the proteins stabilized by the presence of e32, we coupled YFP-SA2 isoform CLIP with Mass Spectrometry. Three biological replicate IPs were prepared from HCT116OsTIR cells that were transfected with either YFP-SA2$^{e32+}$ or YFP-SA2$^{e32\Delta}$. YFP IP efficiency for SA2$^{e32+}$ or SA2$^{e32\Delta}$ was similar and both isoforms interacted with core cohesin subunits (*Figure 5—figure supplement 1c* and *Figure 5—figure supplement 1—source data 2*). We identified a total of 238 proteins, the majority of which overlap in the two SA IPs and with a previously published SA2 IP (*Kim et al., 2019*; *Figure 5—figure supplement 1c, d*). We used a pairwise analysis of SA2$^{e32+}$ vs SA2$^{e32\Delta}$ samples to generate a fold-change value for each putative interactor (*Figure 5e* and *Figure 5—source data 2*). GO analysis of proteins changed by at least 1.5-fold, and absent in Mock IP revealed a mild enrichment for post-translational modification category from the SA2$^{e32\Delta}$ IP (FDR = 0.0234, p=1.35e-06), and conversely an enrichment of the RNA Binding category from SA2$^{e32+}$ (FDR = 3.43E-05, p=6.56E-09). Interestingly, the enriched proteins included YTHDC1 and YTHDF3 (previously identified in the SA1$^{\Delta Coh}$ interactome, *Figure 2b*), DIS3 and POLR2B, all known to play key roles in RNA-protein complexes and stability, have molecular weights ~110–140 kDa and thus likely represent the specifically enriched band in the CLIP experiments (*Figure 5d*, black arrow). Finally, the observation that a basic exon 32 domain in SA2 supports the stability of RNA-RBP interactions led us to investigate if exon 32 also stabilized SA2 at R-loops. We repeated the S9.6 IP in RAD21$^{mAC}$ cells expressing either YFP-SA2$^{e32+}$ or YFP-SA2$^{e32\Delta}$. As before, AQR and MCM3 were enriched by S9.6 IP (*Figure 5f*) and we found that SA2$^{e32+}$ was more enriched in the S9.6 IP compared to SA2$^{e32\Delta}$ (enrichment of 1.8-fold and 1.24-fold respectively, relative to endogenous SA2; *Figure 5f and g*). Taken together, our results support a role for the alternatively spliced C-terminal basic domain of SA in stabilizing interactions with RBPs and R-loops.

## Discussion

Whether SA proteins function in their own right outside of the cohesin complex is rarely considered. Consequently, our understanding of how these proteins contribute to cohesin function and disease is incomplete. In this study, we shed light on this question by uncovering a diverse repertoire of SA1 interactors in cells acutely depleted of the cohesin ring. This ranges from proteins associated with

translation and ribosome biogenesis to RNA processing factors and regulators of the epitranscriptome. These observations suggest that SA1 has a previously unappreciated role in post-transcriptional regulation of gene expression which offers much-needed new insight into its roles in disease and cancer.

Acute depletion of the cohesin ring has allowed us to capture a moment in the normal life cycle of cohesin–DNA associations and unveiled a previously unknown step for SA proteins herein. We show that SA proteins, independent of the cohesin trimer, bind to DNA and RNA, either in the context of RNA:DNA hybrid structures, as we have shown here, or perhaps sequentially, and facilitate the loading of cohesin. Future work investigating the binding profiles of SA proteins and re-loaded cohesin following R-loop depletion in vivo, would help to confirm these findings. Furthermore, biochemical assessment of SA interaction with R-loops will help identify the mechanisms of SA's roles at R-loops both in the presence or absence of cohesin rings.

Our results are supportive of biophysical observations of SA proteins and R-loops (*Pan et al., 2020*) and in vitro assessment of cohesin loading at DNA intermediates (*Murayama et al., 2018*). These results point to the importance of DNA *structure,* as opposed to sequence, in the targeting of cohesin to chromatin. Furthermore, structural studies suggest that NIPBL and SA1 together bend DNA and cohesin to guide DNA entering into the cohesin ring (*Shi et al., 2020*; *Higashi et al., 2020*; *Chao et al., 2015*). Our work shows that in cells lacking either the canonical NIPBL/MAU2 loader complex or the SA proteins, cohesin can still associate with chromatin, suggesting that loading can occur with either component alone, albeit most effectively together.

Our results represent a new view onto the role of SA proteins in cohesin biology. Since SA paralogs have distinct terminal ends and nucleic acid targeting mechanisms (*Lin et al., 2016*; *Countryman et al., 2018*), their recruitment to chromatin may be specified by unique DNA, RNA or protein-interactions, or indeed all three. Such diversification of loading platforms would be important in large mammalian genomes to ensure sufficient amounts of cohesin on chromatin or to stabilize cell-type-specific chromatin structures (*Pezic et al., 2023*). Indeed, SA1 and SA2 show clear differences in interaction with F/YXF-motif containing proteins, despite the fact that both paralogs contain a CES domain (*Li et al., 2020*), underscoring the importance of investigating interactions in vivo and arguing that additional features play an important role in complex stabilization. In this context, RNA-associated protein interaction has previously been shown to support cohesin stabilization at CTCF at the *IGF2/H19* locus (*Yao et al., 2010*). These results are in line with our findings that a basic domain in the unstructured C-terminal portion of SA supports RNA-associated protein interactions.

Our study suggests that SA1 may act as a novel regulator of R-loop homeostasis. It is noteworthy that other suppressors of R-loop formation include RNA processing factors, chromatin remodellers and DNA repair proteins *García-Muse and Aguilera, 2019* which all function in the context of nuclear

**Table 2.** siRNAs used in this study.

| siRNA name | Company | Target | Catalogue no. | custom siRNA sequence |
|---|---|---|---|---|
| si scramble control | Dharmacon | Smartpool | D-001810-10-05 | |
| siSA1 | Dharmacon | Smartpool | L-010638-01-0010 | |
| siSA2 | Dharmacon | Smartpool | L-021351-00-0010 | |
| siNIPBL | Dharmacon | Smartpool | L-012980-00-0010 | |
| siAQR | Dharmacon | Smartpool | L-022214-01-0005 | |
| esi control | Sigma | Luciferase | EHUFLUC | |
| esi SA1 | Sigma | exon 31 | custom esiRNA | TCCTCAGATGCAGATCTCTTGGTTAGGCC AGCCGAAGTTAGAAGACTTAAATCGGAAG GACAGAACAGGAATGAACTACATGAAAGTG AGAACTGGAGTGAGGCATGCTGT |
| esi SA2 | Sigma | exon 32 | custom esiRNA | CACGCAGGTAACATGGATGTTAGCTCAAAG ACAACAAGAGGAAGCAAGGCAACAGCAGG AGAGAGCAGCAATGAGCTATGTTAAACTG CGAACTAATCTTCAGCATGCCAT |

bodies (*Misteli, 2007*). We find that SA1 proteins are enriched at very distal chromatin interactions in cohesin-depleted Hi-C data, interact with numerous RBPs known to condense in 3D (*Nozawa et al., 2017*; *Huo et al., 2020*) and are enriched in S9.6 domains in cells where we find cohesin becomes associated with chromatin. Harnessing such condensates would provide an efficient loading platform for cohesin at sites of similar biological function. Yeast cohesin has been shown to mediate phase separated condensate structures (*Ryu et al., 2020*). Our results support this view and further suggest that it is SA (and possibly predominantly SA1 in HCT116 cells), with its propensity for intrinsically disordered domains (*Pezic et al., 2023*) that contribute to this formation, thereby linking cohesin loading to biological functions. We note that if SA paralogs or isoforms direct different localization of cohesin loading or stability of its association, this could have important implications in our understanding of disease where SA proteins are commonly mutated, such as in cancers.

## Methods
### Cell culture and IAA-mediated degradation of Rad21
HCT116 cells with engineered RAD21-miniAID-mClover (RAD21mAC), or OsTIR1-only, or both (RAD21mAC-OsTIR) were obtained from Masato T. Kanemaki. Throughout this study we used RAD21mAC-OsTIR cells, and for simplicity we refer to them in the text as RAD21$^{mAC}$. The cells were tested for mycoplasma at the beginning of the project. The cells were maintained in McCoy's 5 A medium with Glutamax (Thermo Fisher Scientific) supplemented with 10% Heat-inactivated FBS (Gibco), 700 µg/ml Geneticin, 100 µg/ml Hygromycin B Gold and 100 µg/ml Puromycin as described. We clonally selected the RAD21mAC-OsTIR cells by sorting green-fluorescence-positive single cells on a FACS Aria Fusion cell sorter (BD Bioscience). Single cells were individually seeded into one well of a 96-well plate, expanded for 10 days into 6 cm culture dishes and selected with Geneticin, Hygromycin B Gold, and Puromycin as indicated above in McCoy's medium for another 10 days. Each clone was assessed for efficiency of Rad21 degradation using FACS analysis and western blotting (WB) using mClover, mAID and OsTIR antibodies. Two clones (H2 and H11) were taken forward and used throughout this study. To deplete RAD21, RAD21mAC-OsTIR cells were grown in adherent conditions for 3 days and treated with 500 µM Indole-3-acetic acid (IAA, Auxin, diluted in EtOH) for 4 hr. For IAA withdrawal, IAA-treated cells were washed with PBS and replaced with fresh supplemented McCoy's medium for another 4 hr. Cells were washed twice with ice-cold PBS before being harvested for later experimental procedures.

### siRNA-mediated knockdowns
For siRNA transfections, RAD21$^{mAC}$-OsTIR cells were reverse transfected with scramble siRNA (siCon) or siRNAs targeting SA1, SA2, NIPBL, or AQR (Dharmacon, Horizon Discovery). A final concentration of 10 nM of siSA1, siSA2, or siNIPBL or 5 nM of siAQR was reverse transfected into the cells using Lipofectamine RNAiMAX reagent (Invitrogen), as per the manufacturer's instructions. Cells were plated at a density of 1–1.25 x 10$^6$ cells per 10 cm dish and harvested 72 hr post-transfection, at a confluency of ~70%. The Lipofectamine-containing media was replaced with fresh media 12–16 hr post-transfection to avoid toxicity. For *Figure 5f/g*, incubation time was reduced to 40 hr. To account for the reduced growth time, cells were plated at a density of 2–3 x 10$^6$ cells per 10 cm dish. Here siCon- and siNIPBL-transfected cells were plated at a lower cell number than siAQR-transfected cells to ensure equalized confluence (~70%) at the time of collection. When IAA-treatment was combined with siRNA-mediated KD, the IAA was added at the end of the normal KD condition so that total KD time was not changed compared to UT cells. For esiRNA treatment, RAD21$^{mAC}$-OsTIR cells were reverse-transfected with 20 µM FLUC control esiRNA or esiRNA custom designed to SA1 exon31 or SA2 exon32 (MISSION siRNA, Sigma Aldrich) using RNAiMAX (Invitrogen). Cells were incubated in transfection mixture for 7–8 hr before being replaced with fresh supplemented McCoy's medium and left for another 40 h until harvest. Efficiency of KD was assessed by WB. siRNA information can be found in *Table 2*.

### Immunofluorescence
Cells were adhered onto poly L-lysine-coated glass coverslips in six-well culture dishes and were washed twice with ice-cold PBS before IF procedures. For RAD21-depletion analysis, cells were fixed

for 10 min at room temperature with 3.7% paraformaldehyde (Alfa Aesar) in PBS, washed three times with PBS and then permeabilized at room temperature for 10 min with 0.25% Triton X-100 in PBS (Sigma Aldrich). For R-loop imaging, cells were fixed and permeabilized with ice-cold ultra-pure mEtOH (Sigma Aldrich) for 10 min at –20 °C. After three washes with PBS, cells were blocked for 45 min at room temperature with 10% FCS-PBS. For RNASEH1 enzyme treatment, cells were incubated with blocking solution supplemented with 1 x RNASEH1 reaction buffer alone (50 mM Tris-HCl, 75 mM KCl, 3 mM MgCl$_2$, 10 mM DTT) or 5 units of RNASEH1 enzyme (M0297, New England Biolabs) for 30 min at 37 °C, PBS-washed twice, before blocking. Cells were washed twice with PBS before incubation with primary antibodies diluted in 5% FCS-PBS at 4 °C overnight. Anti-SA1, anti-SA2 and anti-AQR were used at 1:3000 dilutions; anti-CTCF was used at 1:2500 dilution; anti-s9.6 was used at 1:1000 dilution; anti-V5 was used at 1:1000. After four washes with PBS, cells were incubated with secondary antibodies (donkey anti-Goat AF555 or AF647 for SA1/2 used at 1:3000; donkey anti-Rabbit AF647 for CTCF used at 1:2500; donkey anti-Mouse AF555 for s9.6 used at 1:2000; donkey anti-Rabbit AF647 for AQR used at 1:3000; donkey anti-Rabbit AF488 for V5 used at 1:2000) in 5% FCS-PBS for 1 hr at room temperature, and washed 4 times with PBS before being mounted onto glass slides with ProLong Diamond Antifade Mountant with DAPI (Thermo Fisher Scientific) to stabilize overnight in dark before imaging. See *Table 3* for details of where antibodies were purchased.

Imaging was performed on Zeiss LSM confocal microscopes using 63 x/1.40 NA Oil Plan-Apochromat objective lens (Carl Zeiss, Inc). Images were captured as z-stacks and under consistent digital gain, laser intensity and resolution for each experiment. Numerical analysis was carried out using Imaris software (Oxford Instruments, version 9.5.1) and representative images are shown as maximum z-projected views generated using Fiji Image J. In brief, z-stack images were imported into Imaris, cells were identified using DAPI and only those located 1 μm away from image boundary and sized between 120 and 800 μm$^3$ were selected. A seed-split function of 7.5 μm was used to separate closely situated cells. Fluorescence intensities of individual DAPI-selections in each channel were determined by Imaris and exported into Excel for further analysis. Distribution plots were generated from >50 cells of each replicate with three biological replicates per experiment. Student's *t*-test was performed between control and experimental conditions and statistical significance was determined by detecting the difference between means (unequal variance, two-tailed). Significance is denoted as $p > 0.05$ = not significant (ns), $p \leq 0.05$ = *, $p \leq 0.005$ = **, $p \leq 0.0005$ = *** and $p \leq 0.0001$ = ****.

## Chromatin fractionation and co-Immunoprecipitation

Cells were washed twice with ice-cold PBS (Sigma Aldrich) and lysed in Buffer A (10 mM HEPES, 10 mM KCl, 1.5 mM MgCl2, 0.34 M Sucrose, 10% Glycerol, 1 mM DTT, 1 mM PMSF/Pefabloc, protease inhibitor), supplemented with 0.1% T-X100, for 10 min on ice. Lysed cells were collected by scraping. Nuclei and cytoplasmic material was separated by centrifugation for 4 min at 1300 *g* at 4 oC. The supernatant was collected as the cytoplasmic fraction and cleared of any insoluble material with further centrifugation for 15 min at 20,000 *g* at 4 °C. The nuclear pellet was washed once with buffer A before lysis in buffer B (3 mM EDTA, 0.2 mM EGTA, 1 mM DTT, 1 mM PMSF/Pefabloc, protease inhibitor) with rotation for 30 min at 4 oC. Insoluble nuclear material was spun down for 4 min at 1700 *g* at 4 °C and the supernatant taken as nuclear soluble fraction. The insoluble material was wash once with buffer B and then resuspended in high-salt chromatin solubilization buffer (50 mM Tris-HCl pH 7.5, 1.5 mM MgCl2, 500 mM KCl, 1 mM EDTA, 20% Glycerol, 0.1% NP-40, 1 mM PMSF/Pefabloc, protease inhibitor). The lysate was vortexed for 2 min to aid solubilization. Nucleic acids were digested with 85 U benzonase (Sigma-Aldrich) per 100x106 cells, with incubation for 10 min at 37 °C and 20 min at 4 °C. Chromatin was further solubilized with ultra-sonication for 3x10 s at an amplitude of 30. The lysate was diluted to 200 mM KCl and insoluble material was removed by centrifugation at 15,000 RPM for 30 min at 4 oC.

For coIP, antibodies were bound to Dynabead Protein A/G beads (ThermoFisher Scientific) for 10 min at room temperature and ~5 hr at 4 °C. For mock IgG IPs, beads were incubated with serum from the same host type as the antibody of interest. One mg of chromatin extract was incubated with the antibody-bead conjugate per IP for approximately 16 hr at 4 °C. IPs were washed x5 with IP buffer (200 mM chromatin solubilization buffer) and eluted by boiling in either 2 x Laemmli sample buffer (BioRad) or 4 x NuPAGE LDS sample buffer (ThermoFisher Scientific). Proteins ≤250 kDa were separated by SDS-PAGE electrophoresis using 4–20% Mini-PROTEAN TGX Precast Protein Gels (BioRad) and transferred to Immobilon-P PVDF Membrane (Merck Millipore) for detection. Proteins ≥250 kDa

**Table 3.** Antibodies used in this study.

| Protein | Company | Catalogue No. | Species | Main Figure Reference |
|---|---|---|---|---|
| SA1 | Abcam | ab4455 | Mouse | 1 a, c, d, e, 2 a, b, d, 3 a, b, c, d, e, g, h 4f,g, 5b, c, f, h, i |
| SA1 | Abcam | ab4457 | Mouse | 1i |
| SA2 | Bethyl | A300-159 | Goat | 1b, c, d, , 2 a, 3 a, b, c, f, 4 g, e, f, 5 c, d, e, f, g |
| SA2 | Bethyl, AbVantage Pack | A310-941A | Goat | 1i |
| CTCF | Diagenode | C15410210 | Rabbit | 1 c, d, i, 2 a |
| CTCF | Cell signalling | 2899 s | Rabbit | 1 a, e |
| RAD21 | Abcam | ab992 | Rabbit | 1 c, d, i, 2 a, d, 4d, e, g, j |
| GFP-TRAP | ChromoTek | gtd-20 | | 1i, 5f |
| GFP | Invitrogen | A11122 | Rabbit | 1 a, e |
| mAID | MBL | M214-3 | Mouse | *Figure 1—figure supplement 1a* |
| OsTIR | MBL | PD048 | Rabbit | *Figure 1—figure supplement 1a* |
| SMC3 | Abcam | ab9263 | Rabbit | 1i |
| CHD6 | Bethyl | A301-221A | Rabbit | 2 a |
| MCM3 | Bethyl | A300-124A | Goat | 2 a, 3g, 5f |
| HNRNPUL2 | Abcam | ab195338 | Rabbit | 2 a |
| YTHDC1 | Abcam | ab122340 | Rabbit | 2d |
| FTSJ3 | Bethyl | A304-199A-M | Rabbit | 2d |
| FANCI | Bethyl | A301-254A-M | Rabbit | 2d |
| TAF15 | Abcam | ab134916 | Rabbit | 2d |
| DHX9 | Abcam | Ab26271 | Rabbit | 2d |
| SSRP1 | Abcam | ab26212 | Mouse | 2d |
| INO80 | Proteintech | 18810–1-AP | Rabbit | 2d |
| ESYT2 | Sigma-Aldrich | HPA002132 | Rabbit | 2d |
| S9.6 | Kerafast | ENH001 | Mouse | 3d, g, h, 5f |
| RNASE H2 | Novus | NBP1-76981 | Rabbit | 3g |
| AQR | Bethyl | A302-547A | Rabbit | 3g, 5f, e |
| POLR2 | Covance | MMS-1289 | Mouse | 3g |
| MAU2 | Abcam | ab183033 | Rabbit | 4d, f |
| NIPBL | Abbiotec | 250133 | Rat | 4d, f, |
| H3 | Abcam | ab1791 | Rabbit | 4d, f, 5c |

| Name (SecondaryAbs) | Fluorophore | Company | Catalogue No. | Figure Reference |
|---|---|---|---|---|
| Donkey anti-Rabbit | Cy3_AF647 | Home made from Jackson Immunoresearch IgG | Home made from 711-005-152 | 1e, *Figure 1—figure supplement 1* |
| Donkey anti-Goat | AF405_AF647 | Home made from Jackson Immunoresearch IgG | Home made from 705-005-147 | 1e, *Figure 1—figure supplement 1* |
| Donkey anti-mouse | AF647 | Invitrogen | A31570 | 1a, e, *Figure 1—figure supplement 1b, e, Figure 3—figure supplement 1f* |

*Table 3 continued on next page*

*Table 3 continued*

| Name (SecondaryAbs) | Fluorophore | Company | Catalogue No. | Figure Reference |
|---|---|---|---|---|
| Donkey anti-rabbit | AF488 | Invitrogen | A21206 | 1a, e, *Figure 1—figure supplement 1b, e*, *Figure 3—figure supplement 1f* |
| Donkey anti-rabbit | AF647 | Invitrogen | A31573 | 1a, e, *Figure 1—figure supplement 1b, e*, *Figure 3—figure supplement 1f* |
| Donkey anti-goat | AF555 | Invitrogen | A21432 | 1a, e, *Figure 1—figure supplement 1b, e*, *Figure 3—figure supplement 1f* |
| Donkey anti-goat | AF647 | Invitrogen | A21447 | 1a, e, *Figure 1—figure supplement 1b, e*, *Figure 3—figure supplement 1f* |
| Goat anti-Mouse | AF568 | ThermoFisher Scientific | A-11031 | 1a, e, *Figure 1—figure supplement 1b, e*, *Figure 3—figure supplement 1f* |
| Goat anti-Rabbit | AF647 | ThermoFisher Scientific | A-21244 | 1a, e, *Figure 1—figure supplement 1b, e*, *Figure 3—figure supplement 1f* |
| Rabbit anti- Goat | AF647 | ThermoFisher Scientific | A-21446 | |

were separated by SDS-PAGE electrophoresis using Invitrogen NuPAGE 3–8% Tris-Acetate precast protein gels. Transfer was extended to overnight with low voltage (20 V) to aid in transfer of the high-molecular-weight proteins. Membranes were incubated in primary antibody solution overnight at 4 °C and images were detected using chemiluminescent fluorescence. Densitometry was carried out using ImageStudio Lite software with statistical significance calculated by unpaired t test, unless otherwise specified. Fold enrichment quantifications were performed by first normalizing the raw densitometry value to its corresponding Histone H3 quantification and the comparing between the samples indicated. See *Table 3* for details of antibodies.

## S9.6 IP and Dot Blot

Cells were fractioned and processed for S9.6 IP as described above, with the following modifications. To avoid digestion of RNA:DNA hybrids, samples were not treated with benzonase during chromatin solubilization and sonication was carried out for 10 min (Diagenode Biorupter) as in *Cristini et al., 2018*. Where indicated, chromatin samples were treated with Ribonuclease H enzyme (NEB) overnight at 37 °C to digest RNA:DNA hybrids in the extract. To avoid detection of single-stranded RNA by the S9.6 antibody, all S9.6 IP samples were pre-treated with Purelink RNase A (Thermo Fisher Scientific) at 0.25 µg/1 mg chromatin extract for 1 hr 30 min at 4 °C. The reaction was stopped with addition of 143 U Invitrogen SUPERase•In RNase Inhibitor (Thermo Fisher Scientific). RNA:DNA hybrid levels were assessed in chromatin samples by dot blot. Specifically, the chromatin lysate was directly wicked onto Amersham Protran nitrocellulose membrane (Merck) by pipetting small volumes above the membrane. Membranes were blocked in 5% (w/v) non-fat dry milk in PBS-0.1% Tween and incubated with S9.6 antibody overnight as for standard western blot. As above, detection was carried out using chemiluminescent fluorescence. RNase A-mediated digestion of RNA:DNA hybrids was performed using a non-ssRNA-specific enzyme (Thermo Scientific) at 1.5 µg/25 µg chromatin extract at 37 °C.

## ChIP-sequencing, library preparation, and analysis

ChIP lysates were prepared from RAD21^mAC cells treated with EtOH or IAA for 6 hr in two biological replicates. Formaldehyde (1%) was added to the culture medium for 10 min at room temperature. Fixation was blocked with 0.125 M glycine and cells were washed in cold PBS. Nuclear extracts were prepared by douncing (20 strokes, medium pestle) in swelling buffer (25 mM HEPES pH8, 1.5 mM MgCl2, 10 mM KCL, 0.1% NP40, 1 mM DTT and protease inhibitors) and centrifuged for 5 min at 2000 rpm at 4 C. Nuclear pellets were resuspended in Sucrose buffer I (15 mM Hepes pH 8, 340 mM Sucrose, 60 mM KCL, 2 mM EDTA, 0.5 mM EGTA, 0.5% BSA, 0.5 mM DTT and protease inhibitors) and dounced again with 20 strokes. The lysate was carefully laid on top of an equal volume of Sucrose buffer II (15 mM Hepes pH 8, 30% Sucrose, 60 mM KCL, 2 mM EDTA, 0.5 mM EGTA, 0.5 mM DTT

and protease inhibitors) and centrifuged for 15 min at 4000 rpm at 4 °C. Nuclei were washed twice to remove cytoplasmic proteins, centrifuged and the pellet was resuspended in Sonication/RIPA buffer (50 mM Tris, pH 8.0, 140 mM NaCl, 1 mM EDTA, 1% Triton X-100, 0.1% Na-deoxycholate, 0.1% SDS and protease inhibitors) at a concentration of $5x10^6$ nuclei in 130 µl buffer. This was transferred to a sonication tube (AFA Fiber Pre-Slit Snap-Cap 6x16 mm) and sonicated in a Covaris S2 (settings; 4 cycles of 60 s, 10% duty cycle, intensity: 5, 200 cycles per burst). Soluable chromatin was in the range of 200–400 bp. Triton X100 was added (final concentration 1%) to the sonicated chromatin and moved to a low-retention tube (Eppendorf) before centrifugation at 14,000 rpm for 15 min at 4 °C and pellets were discarded. 1/100th of the chromatin lysate was retained as the Input sample.

For Immunoprecipitation, 200 µg chromatin aliquots/IP were precleared with a slurry of Protein A/G (50:50) (Dynabeads) an incubated for 4 hr at 4 °C. Meanwhile, washed Protein A/G beads (40 µl per IP) were mixed with primary antibodies and incubated for 4 hr at 4 °C. The following amounts of antibodies were used: anti-CTCF, 5 µg/ChIP; anti-SA1, 15 µg/ChIP; anti-SA2, 10 µg of the mixed antibody pack/ChIP; anti-Smc3, 5 µg/ChIP and anti-IgG, 10 µg/ChIP. See *Table 3* for information about the antibodies. Washed, pre-bound Protein A/G beads +antibody were mixed with pre-cleared chromatin lysates and incubated overnight with rotation at 4 °C. The next day, the supernatant was removed and the beads were washed nine times with increasing salt concentrations. Protein-DNA crosslinks were reversed in ChIP elution buffer (1% SDS, 5 mM EDTA, 10 mM Tris HCl pH 8)+2.5 µl of Proteinse K and incubated for 1 hr at 55 °C and overnight at 65 °C. Samples were phenol–chloroform extracted, resuspended in TE buffer and assessed by qPCR as a quality control. Libraries were prepared from 5 to 10 ng of purified DNA, depending on availability of material, using NEBNext Ultra II DNA Library Prep Kit for Illumina kit and using NEBNext Multiplex Oligos for Illumina (Index Primers Set 2) according to manufacturer's instructions using 6–8 cycles of PCR. ChIP-seq libraries from one biological set (all ChIP libraries for both EtOH and IAA) were multiplexed and sequenced on the Illumina HiSeq2500 platform, 80 bp single-end reads. Each biological set was sequenced on a separate run.

Quality control of reads was preformed using FASTQC. Reads were aligned to the hg19 reference genome using Bowtie with three mismatches. PCR duplicates were detected and removed using SAMTOOLS. Bam files were imported into MISHA (v 3.5.6) and peaks were identified using a 0.995 percentile. Peaks that overlapped in both replicates were retained. Only replicate 1 of the SA1 library was used. Correlation plots of peaks across the genome from different ChIP libraries were compared with log-transformed percentiles plotted as a smoothed scatter plot. Comparison of peaks at regions of interest were carried out using deepTools (Version 3.1.0–2). For input into deepTools, peak data was converted to bigwig format, with a bin size of 500, using the UCSC bedGraphtoBigWig package. The signal matrix was calculated for a window 2000 bp up- and down-stream of the region of interest, missing data was treated as zero, and all other parameters were as default. Heatmaps were generated within deepTools, with parameters as default. Read density profile plots were plotted in ggplot using deepTools profilePlot -perGroup data and smoothed using geom_smooth default 'gam' settings.

## DRIP-sequencing

DRIP lysates were prepared from chromatin. Chromatin was fractionated as described for ChIP samples above, with the following changes. Samples were not fixed and were collected from the plate by scraping in ice-cold PBS. Sonication was performed to solubilize the chromatin using a picorupter with 10 cycles of 30 s on, 30 s off. Following sonication, 60 U of RNase I (Ambion) was incubated with each sample for 1.5 hr at 37 °C to reduce off-target RNA binding by S9.6. Protein was digested with 184 µg of proteinase K incubated with each sample for 2.5 hr at 45 °C and 3.5 hr at 55 °C. Samples were spun briefly and the supernatant taken. Nucleic acid material was isolated from the sample by phenol-chloroform extraction and isopropanol precipitation. Purified nucleic acid material was IP'd as in *Skourti-Stathaki et al., 2019*, including resuspension nuclear lysis buffer and fragmentation by sonication using a picorupter for 4 cycles of 30 s on, 30 s off. Following elution, reverse crosslinking, and proteinase K treatment the immunoprecipitated nucleic acid was purified by phenol-chloroform

**Table 4.** Published ChIP-seq datasets used for ChromHMM.

| Protein | Accession no. | Publication | Matched input |
|---|---|---|---|
| NIPBL (EtOH- and IAA-treated) | GSE104334 | *Rao et al., 2017* | - |
| CBX1 | GSM1010758 | *Gertz et al., 2013* | |
| EZH2 | GSM3498250 | *Dunham et al., 2012* | GSM2308475;GSM2308476 |
| POLR2A | GSM935426 | *Dunham et al., 2012* | GSM2308422 |
| POLR2AphosphoS5 | GSM803474 | *Gertz et al., 2013* | GSM803475 |
| SIN3A | GSM1010905 | *Gertz et al., 2013* | |
| YY1 | GSM803354 | *Gertz et al., 2013* | GSM803475 |
| H3K4me1 | GSM945858 | | GSM2308475; GSM2308476 |
| H3K4me1 | GSM2527549 | *Dunham et al., 2012* | GSM2308422 |
| H3K4me3 | GSM2533929 | *Dunham et al., 2012* | GSM2308475; GSM2308476 |
| H3K4me3 | GSM945304 | *Thurman et al., 2012* | GSM945287 |
| H3K9me3 | GSM2527565 | *Dunham et al., 2012* | |
| H3K9me3 | GSM2308431 | *Dunham et al., 2012* | |
| H3K27ac | GSM2534277 | *Dunham et al., 2012* | GSM2308422 |
| H3K27me3 | GSM2308612 | *Dunham et al., 2012* | |

extraction and isopropanol precipitation. Second strand synthesis was carried out according to *Nadel et al., 2015* with minor changes. Namely, the reaction was set up in a PCR tube as:~28 µl eluted DRIP sample, ~27 µl nuclease-free water (depending on DRIP sample, for a total of 75 µl), 15 µl 5 X ss buffer, 2 µl 10 mM dNTP, 0.5 µl DNA ligase (*E. coli*, NEB cat. M0205S), 2 µl DNA polymerase I (*E. coli*, NEB cat. M0209S), 0.5 µl RNase H (2.5 units). This was incubated for 2 hr at 16 °C. dsDNA was purified using a 1.8 X ratio of SPRI beads and eluted in ultrapure water. Libraries were prepared using NEB Ultra II DNA library prep kit according to manufacturer's instructions and sequenced on the Illumina NovaSeq platform, 100 bp paired-end reads (30 M per sample). Reads were aligned and processed as for the ChIP-seq samples above with two changes; (i) reads were quality trimmed using trimfq -b 14 -e 16, and (ii) a maximum insert size of 1000 was set for bowtie. DeepTools analysis was carried out as for ChIP-seq samples and with binSize set to 2 bp.

## ChromHMM
ChIP-seq data for YY1, CBX3, SIN3A, POLR2A, POLR2AphosphoS5, H3K27ac, H3K4me3, H3K4me1, H3K27me3, EZH2, and H3K9me3 from HCT116 cells were obtained from ENCODE and processed as above (see *Table 4* for references). ChIP-seq, DRIP-seq, and ENCODE ChIP-seq. BAM files were binarized in ChromHMM using a bin size of 200 bp and a shift of 150 bp. Where input files were available on ENCODE they were used in ChromHMM to determine the binarization threshold, otherwise the ChromHMM default of a uniform background was assumed. The chromatin state model was generated for 15 states and compared to the hg19 genome assembly. All other parameters were as default.

## Hi-C data and contact hotspots analysis
Generating hotspots - Previously published Hi-C datasets derived from RAD21[mAC] cells treated with EtOH or IAA (*Rao et al., 2017*) were analyzed as previously described *Barrington et al., 2019*. Custom R scripts were written to identify Hi-C hotspots, that is regions of Hi-C maps with high contact frequency. To begin, for each chromosome, all contacts were extracted and subsetted for only high scoring ( ≥ 60) contacts between a band of 10e3 – 70e6. Using KNN, for each high scoring contact, the 250 nearest neighbour contacts were identified and subset for only the high-scoring neighbours. This created a list of high scoring neighbours for each high scoring contact, where the first neighbour is the contact itself with a distance of 0. This allowed the neighbour information to be converted into edge information, thereby allowing high score fend contacts to be grouped into cluster hotspots using the

R package 'igraph'. Hotspots that contained less than the minimum number of high scoring fends (<100) were removed. The output list of hotspots were represented as 2D intervals which contained high scoring contacts. In total, 5539 hotspots were identified in EtOH and 759 in IAA Hi-C data.

Creating aggregate plots - To calculate and visualize the contact enrichment at hotspots in the EtOH and IAA Hi-C, we used the R package 'shaman'. Firstly, we used the function 'shaman_generate_feature_grid' to calculate the enrichment profile at EtOH and IAA hotspots. Using the weighted centre for each hotspot, represented as a 2D interval we used the function to build grids for the EtOH and IAA hotspots in the HiC data at 3 specific bands, 100 k – 1 MB, 1MB – 5MB, 5MB – 10MB. A range of 250 kb was visualized around the weighted centre. The grid was built by taking all combinations interval1 and interval2 of the EtOH and IAA hotspot centres, with each combination termed a 'window'. Hotspots were not filtered for size or shape. A score threshold of 60 was used to focus on enriched pairs, those windows that did not contain at least one point with a score of 60 were discarded. Each window was then split into 1000nt bins and the windows were summed together to generate a grid containing the observed and expected contacts. We visualized the grid using 'shaman_plot_feature_grid' using 'enrichment' mode and a plot_resolution value of 6000, due to the large range being visualized.

## STORM – Immunolabeling and imaging

Two clones of RAD21$^{mAC}$-OsTIR cells were seeded at a density of 30,000 cells per well per 400 ul onto poly-L-lysine-coated 8-well chamber slides (Lab-Tek 155411) overnight. Each clone was treated with EtOH, IAA, or IAA washoff and then fixed with PFA 4% (Alfa Aesar) for 10 min at room temperature and rinsed with PBS three times for 5 min each. The cells were shipped to the Cosma Lab after fixation for STORM processing and imaging. Cells were permeabilized with 0.3% Triton X-100 in PBS and blocked in blocking buffer (10% BSA – 0.01% Triton X-100 in PBS) for 1 hr at room temperature. Cells were incubated with primary antibodies (see *Table 3*) in blocking buffer at 1:50 dilution. For combined S9.6/*STAG1* and S9.6/RAD21-GFP imaging, cells were incubated with primary antibodies in blocking buffer (dilutions 1:100 for S9.6 and *STAG1*, 1:250 for GFP). Cells were washed three times for 5 min each with wash buffer (2% BSA – 0.01% Triton X-100 in PBS) and incubated in secondary antibody. For STORM imaging, home-made (*Bates et al., 2007*) dye pair labeled secondary antibodies were added at a 1:50 dilution in blocking buffer and were incubated for 45 min at room temperature or single fluorophore labeled commercial antibodies were added at a 1:250 dilution in blocking buffer and were incubated for 45 min at room temperature (see *Table 3*). Cells were washed three times for 5 min each with wash buffer.

STORM imaging was performed on an N-STORM 4.0 microscope (Nikon) equipped with a CFI HP Apochromat TIRF 100x1.49 oil objective and an iXon Ultra 897 camera (Andor) and using Highly Inclined and Laminated Optical sheet illumination (HILO). Dual color STORM imaging was performed with a double activator and single reporter strategy by combining AF405_AF647 anti-Goat secondary with Cy3_AF647 anti-Rabbit secondary antibodies. Sequential imaging acquisition was performed (1 frame of 405 nm activation followed by 3 frames of 647 nm reporter and 1 frame of 560 nm activation followed by 3 frames of 647 nm reporter) with 10ms exposure time for 120,000 frames. 647 nm laser was used at constant ~2 kW/cm$^2$ power density and 405 nm and 560 nm laser powers were gradually increased over the imaging. For S9.6 experiments, before STORM imaging, conventional images were taken for s9.6 signal (AF568 labeled) with a TRITC filter, for endogenous mClover signal with a FITC filter and for *STAG1* or GFP (AF647 labeled) with a Quadband filter. STORM imaging for *STAG1* or GFP was performed with continuous imaging acquisition (i.e. simultaneous stimulation with 405 and 647 nm lasers) with 10ms exposure time for 60000 frames. 647 nm laser was used at constant ~2 kW/cm$^2$ power density and 405 nm was gradually increased over the imaging. Imaging buffer composition for STORM imaging was 100 mM Cysteamine MEA (Sigma-Aldrich, #30070)–5% Glucose (Sigma-Aldrich, #G8270) – 1% Glox Solution (0.5 mg/ml glucose oxidase, 40 mg/ml catalase (Sigma-Aldrich, #G2133 and #C100)) in PBS.

## STORM imaging analysis and quantifications

STORM images were analyzed and rendered with Insight3 software (kind gift of *Huang et al., 2008*) as previously described (*Bates et al., 2007*; *Rust et al., 2006*). Localizations were identified based on a threshold and fit to a simple Gaussian to determine the x and y positions. Cluster analysis of CTCF,

SA1 and SA2 STORM signal was performed as previously described (*Ricci et al., 2015*) to obtain cluster size and positions and to measure Nearest Neighbour distributions (NND) between clusters of the same protein in individual nuclei. NND between clusters' centroids of two different proteins (i.e. CTCF-SA1 and CTCF-SA2) was calculated by knnsearch.m Matlab function and the NND histogram of experimental data was obtained by considering all the NNDs of individual nuclei (histogram bin, from 0 to 500 nm, 5 nm steps). Simulated NNDs recapitulating random spatial distribution of cluster centroids were first obtained for each nucleus separately and then merged to calculate the simulated NND histogram (histogram bin, from 0 to 500 nm, 5 nm steps). The difference plot reports the difference between experimental NND and simulated NND. Quantification and analysis of STORM images was performed in Matlab and statistical analysis was performed in Graphpad Prism (v7.0e). The type of statistical test is specified in each case. Statistical significance is represented as indicated above.

Analysis for S9.6 experiments was performed in the following way. After generating localization lists for each STORM image, nuclear masks and S9.6 were generated to segment the obtained localizations belonging to nuclear areas inside or outside s9.6 enriched regions. Masks generation, quantification of masks' areas and segmentation of STORM localizations were performed in Fiji/ImageJ. Nuclear masks were manually designed based on *STAG1*/GFP signal. S9.6 masks were generated by applying an automatic threshold on s9.6 images based on nuclear intensity signal. Masks were visually inspected individually and adjusted manually in cases where dim signal or noise from cytosolic signal compromised the identification of the mask. The area of all masks was calculated. Masks were applied to the STORM localization lists to generate segmented lists with the localizations belonging to the entire nucleus and to the s9.6 enriched areas. Finally, the density of localizations (number of localizations/area of the mask) for the areas inside and outside s9.6 masks was calculated and the ratio between both values was plotted.

Graphpad Prism software used for statistical analysis can be found at: https://www.graphpad.com/scientific-software/prism/ MatLab software used for imaging data analysis can be found at: https://www.mathworks.com/products/matlab.html.

## Mass spectrometry sample preparation and data collection

SA1 immunoprecipitation samples were analysed by liquid chromatography–tandem mass spectrometry (LC-MS/MS). Three biological replicate experiments were carried out for MS and each included four samples, untreated (UT), treated with IAA for 4 hr, siCon, or siSA1, generated as described above. Cells were fractionated to purify chromatin-bound proteins as above and immunoprecipitated with IgG- or SA1-bead conjugates. To maximize IP material for the MS, the antibody amount was increased to 15 µg and the chromatin amount was increased to 2 mg.

For MS analysis of STAG2 isoform interactors, transfection conditions and immunoprecipitations of YFP-STAG2+ex32 and YFP-STAG2Δex32 were prepared as described in 'GFP-TRAP + Cloning of STAG2 isoforms and YFP constructs'. To maximise IP/MS intensity, 95% of whole cell lysates isolated from each condition was used for immunoprecipitation with 40 µL of GFP-Trap slurry in each of the 3 independent biological replicates. Each sample was eluted by boiling in 50µl of 2x Laemmli buffer and processed for MS same as SA1 IP/MS.

For both SA1 IPs and SA2-YFP experiments, the IP eluates were loaded into a pre-cast SDS-PAGE gel (4–20% Mini-PROTEAN TGX Precast Protein Gel, 10-well, 50 µL) and proteins were run approximately 1 cm to prevent protein separation. Protein bands were excised and diced, and proteins were reduced with 5 mM TCEP in 50 mM triethylammonium bicarbonate (TEAB) at 37 °C for 20 min, alkylated with 10 mM 2-chloroacetamide in 50 mM TEAB at ambient temperature for 20 min in the dark. Proteins were then digested with 150 ng trypsin, at 37 °C for 4 hr followed by a second trypsin addition for 4 hr, then overnight at room temperature. After digestion, peptides were extracted with acetonitrile and 50 mM TEAB washes. Samples were evaporated to dryness at 30 °C and resolubilized in 0.1% formic acid.

nLC-MS/MS was performed on a Q Exactive Plus interfaced to a NANOSPRAY FLEX ion source and coupled to an Easy-nLC 1200 (Thermo Scientific). 25% (first, second and fourth biological replicate) or 50% (third biological replicate) of each sample was loaded as 5 or 10 µL injections. Peptides were separated on a 27 cm fused silica emitter, 75 µm diameter, packed in-house with Reprosil-Pur 200 C18-AQ, 2.4 µm resin (Dr. Maisch) using a linear gradient from 5% to 30% acetonitrile/ 0.1% formic acid over 60 min, at a flow rate of 250 nL/min. Peptides were ionized by electrospray ionization using

1.8 kV applied immediately prior to the analytical column via a microtee built into the nanospray source with the ion transfer tube heated to 320 °C and the S-lens set to 60%. Precursor ions were measured in a data-dependent mode in the orbitrap analyser at a resolution of 70,000 and a target value of 3e6 ions. The ten most intense ions from each scan were isolated, fragmented in the HCD cell, and measured in the orbitrap at a resolution of 17,500.

## Mass spectrometry data analysis

Raw data was analysed with MaxQuant (*Cox and Mann, 2008*) version 1.5.5.1 where they were searched against the human UniProtKB database using default settings (http://www.uniprot.org/, downloaded 17/12/2019). For the isoform MS experiment, raw data was analysed with MaxQuant (ref as above ), version 1.6.17 where they were searched against the human UniProtKB database including the STAG2ex32Δ and the YFP sequences (http://www.uniprot.org/, downloaded 06/11/2020). Carbamidomethylation of cysteines was set as fixed modification, and oxidation of methionines and acetylation at protein N-termini were set as variable modifications. Enzyme specificity was set to trypsin with maximally two missed cleavages allowed. To ensure high confidence identifications, PSMs, peptides, and proteins were filtered at a less than 1% false discovery rate (FDR). Label-free quantification in MaxQuant was used with LFQ minimum ratio count set to 2 with 'FastLFQ' (LFQ minimum number of neighbours = 3, and LFQ average number of neighbours = 6) and 'Skip normalization' selected. In Advanced identifications, 'Second peptides' was selected and the 'match between runs' feature was not selected. Statistical protein quantification analysis was done in MSstats (*Choi et al., 2014*) (version 3.14.0) run through RStudio. Contaminants and reverse sequences were removed and data was log2 transformed. To find differential abundant proteins across conditions, paired significance analysis consisting of fitting a statistical model and performing model-based comparison of conditions. The group comparison function was employed to test for differential abundance between conditions. Unadjusted p-values were used to rank the testing results and to define regulated proteins between groups. The sample quantification function was used to obtain model-based protein abundance summarisations across biological replicates.

## Downstream proteomic data analysis

Proteins with peptides discovered in the IgG samples were disregarded from downstream analyses. Significantly depleted/enriched proteins were considered with an absolute log2foldchange >0.58 (1.5-fold change) and a p-value <0.1. SA1 interactome analysis was performed in STRING. The network was generated as a full STRING network with a minimum interaction score of 0.7 required. Over-enrichment of GO biological process and molecular function terms was calculated with the human genome as background. Network analysis of the SA1 interactome in IAA-treated samples was generated from the significantly depleted/enriched proteins, with a minimum interaction score of 0.4 required. Two conditions for functional enrichments were considered; (i) enrichment was calculated with the human genome as background to determine the full SA1 interactome in the absence of cohesin, and (ii) enrichment was calculated with the untreated SA1 interactome as background, to determine the statistical effect of cohesin loss of the SA1 interactome itself. The network developed in (i) was manually rearranged in Cytoscape for visual clarity, enriched categories were visualized using the STRING pie chart function and half of the proteins within each category were subset from the network based on pvalue change between UTR and IAA samples.

For SA2 isoform interactome analysis, proteins identified in GFP-Trap immunoprecipitation of untransfected HCT116 cells were eliminated from further analysis. Of the 457 proteins identified, 238 proteins in addition to the presence of YFP and STAG2 proteins were detected in both tested IP experiments. Significant enrichment/depletion of interactors were considered by relative log2foldchange > 0.58 (1.5-fold change) and a p-value < 0.1. Over-enrichment of GO biological process and molecular function terms for both this study and from Kim et al., 2019 [insert citation] were calculated using STRING with the human genome as background. Significance was considered with p<0.05 and FDR<10E-5.

Over-enrichment of the s9.6 interactome was calculated separately using the hypergeometric distribution for comparison with (*Cristini et al., 2018*; *Wang et al., 2018*). Significance was calculated using the dhyper function in R and multiple testing was corrected for using the p.adjust Benjamini & Hochberg method. To compare with a minimal background protein list, http://www.humanproteomemap.

org was analysed on the Expression Atlas database to determine a list of proteins expressed in one or more of three tissue types corresponding to the cell types used across the different studies.

## SLiMSearch analysis

The SLiMSearch tool http://slim.icr.ac.uk/slimsearch/, with default parameters was used to search the human proteome for additional proteins that contained the FGF-like motif determined in *Li et al., 2018* to predict binding to SA proteins. The motif was input as [PFCAVIYL][FY][GDEN]F.{0,1}[DANE].{0,1}[DE]. Along with CTCF, four proteins found to contain the FGF-like motif, CHD6, MCM3, HNRNPUL2 and ESYT2 were validated for interaction with SA.

## CLIP

Crosslinking immunoprecipitation (CLIP) was performed as previously described *Beltran et al., 2016*. Briefly, mESC or HCT116 cells were irradiated with 0.2 J/cm2 of 254 nm UV light in a Stratalinker 2400 (Stratagene). Cells were lysed in 1 ml of lysis buffer with Complete protease inhibitor (Roche). Lysates were passed through a 27 G needle, 1.6 U DNase Turbo (Thermofisher) per $10^6$ cells and 0.8 (low) or 8 U (high) U RNase I (Ambion) per $10^6$ cells added, and incubated in a thermomixer at 37 °C and 1100 rpm for 3 min. Lysates were then cleared by centrifugation and using Proteus clarification spin column, according to the manufacturer's instructions. Endogenous SA1 and SA2 were immunoprecipitated with 10 µg SA1 and SA2 antibodies or non-specific IgG control (Sigma) conjugated to protein G dynabeads (Dynal) for 4 hr at 4 °C. Tagged SA2 proteins were immunoprecipitated from HCT116 cells 40 hour after transfection with 30 µl GFP-Trap beads. IPs were washed three times with high-salt buffer (containing 1 M NaCl and 1 M urea) and once with PNK buffer and RNA labelled with 8 µl radioactive $^{32}$P-gamma-ATP (Hartmann Analytic) for 5 mins at 37 °C. RNPs were eluted in LDS loading buffer (Invitrogen) and resolved on a 4–12% gradient NuPAGE Bis-Tris gel (Invitrogen) and transferred onto 0.2 µm diameter pore nitrocellulose membrane. After blocking with PBST +milk, membranes were washed and exposed overnight to phosphorimager screen (Fuji) and RNA-$^{32}$P visualized using a Typhoon phosphorimager (GE) and ImageQuant TL (GE). Membranes were then immunoblotted for SA1, SA2, and RAD21 and visualized using an ImageQuantLAS 4000 imager (GE). See *Table 3* for details on antibodies.

## GFP-TRAP +Cloning of *STAG2* isoforms and YFP constructs

SA2 cDNAs were cloned directly from HCT116 cells by PCR using KAPA HiFi HotStart PCR kit (Roche) (Fwd: ATGATAGCAGCTCCAGAAAACCAACTG; Rev: TTAAAACATTGACACTCCAAGAACTGATTCATCC). Two major isoforms were detected, SA2$^{\Delta ex32}$ where exon32 has been spliced out and SA2$^{+ex32}$ where exon 32 has been spliced in. Both SA2 cDNAs were cloned into pENTR/D vector (Invitrogen) and then into an N-terminal YFP-tagged Gateway cloning vector (a kind gift from Prof. Endre Kiss-Toth, University of Sheffield). Sequences were confirmed by restriction enzyme digestion and Sanger sequencing. Recombinant YFP-SA2$^{\Delta ex32}$ or YFP-SA2$^{+ex32}$ were transfected into adherent HCT116 cells for 40 hours before being harvested. Cells were lysed and fractionated as indicated for CLIP. One third of the whole cell lysate was pre-cleared with a 50:50 mixture of protein A/G magnetic beads and GFP-Trap (Chromotek, gtd-20) was pre-blocked with 1 mg/mL ultra-pure BSA (AM2616, Invitrogen) for 2 hr at 4 °C. After blocking, GFP-Trap was washed twice with CLIP lysis buffer and added to pre-cleared lysates to immunoprecipitate proteins for 1 h at 4 °C. Samples were washed in high salt buffer (50 mM Tris-HCL pH 7.4, 1 M NaCl, 1 mM EDTA, 1% NP-40, 0.1% SDS, 0.5% sodium deoxycholate, 1 M Urea) and low salt PNK buffer (20 mM Tris-HCL pH 7.4, 10 mM MgCl$_2$, 0.2% Tween-20), and eluted in 2 x Laemmli buffer (Bio-Rad). Proteins were separated by SDS-PAGE on a 4–20% gradient mini-PROTEAN Precast Gel (Bio-Rad) and transferred onto PVDF membrane for visualization.

## VAST-TOOLS

VAST-TOOLS was used to generate Percent Spliced In (PSI) scores, a statistic which represents how often a particular exon is spliced into a transcript using the ratio between reads which include and exclude said exon. Paired-end RNA-seq datasets were submitted to VAST-TOOLS (v2.1.3) using the Mmu genome . Briefly, reads are split into 50nt words with a 25nt sliding window. The 50nt words are aligned to a reference genome using Bowtie to obtain unmapped reads. These unmapped reads are then aligned to a set of predefined exon-exon junction (EJJ) libraries allowing for the quantification of

alternative exon events. The output was further interrogated using a script which searches all hypothetical EEJ combinations between potential donors and acceptors within *STAG1*. PSI scores could be obtained providing there was at least a single read within the RNAseq data that supported the event, although we only considered events supported by a minimum of 50 reads. Calculated PSI values for each alternatively spliced exon as well as the average PSI reported in the text are shown below. See (*Figure 5—figure supplement 1*) for names of published datasets used in this analysis.

## Acknowledgements

This work was supported by a Senior Research Fellowship from the Wellcome Trust awarded to SH (106985/Z/15/Z) and a CRUK PhD studentship awarded to HP. The Proteomics work was supported by the CRUK–UCL Centre Award [C416/A25145]. The CLIP work was supported by grants from the European Research Council (ERC, 311704) and Worldwide Cancer Research (21-0255), to RGJ. We are grateful to Jernej Ule for his support with DRIP-sequencing and to Julian Zagalak and the CRICK sequencing facility for reagents, advice and assistance. We thank Stanimir Dulev for his contributions at the early stages of the project and Jiten Manji for his support with microscopy. We also thank Konstantina Skourti-Stathaki for advice about S9.6 antibody, IFs and R-loops. We are grateful to the members of the Hadjur lab for critical discussions and reading of the manuscript.

## Additional information

### Funding

| Funder | Grant reference number | Author |
| --- | --- | --- |
| Wellcome Trust | 106985/Z/15/Z | Suzana Hadjur |
| Cancer Research UK | PhD studentship | Hayley Porter |

The funders had no role in study design, data collection and interpretation, or the decision to submit the work for publication. For the purpose of Open Access, the authors have applied a CC BY public copyright license to any Author Accepted Manuscript version arising from this submission.

### Author contributions

Hayley Porter, Conceptualization, Data curation, Formal analysis, Investigation, Methodology, Validation, Writing – review and editing, Designed and performed the coIP, Mass spectrometry, DRIP-sequencing and cohesin re-loading experiments, analysed the ChIP, DRIP and Hi-C data and performed the statistical analysis for mass spectrometry, Prepared cellular materials for CLIP, Formatted select figures and edited the manuscript with input from all authors; Yang Li, Data curation, Formal analysis, Investigation, Methodology, Writing – review and editing, Performed and analysed all imaging experiments (apart from STORM), derived clonal lines of RAD21mAC cells, cloned YFP-tagged SA2 cDNAs, and performed CLIP with MTC and CLIP-MS with AB, Prepared cellular materials for CLIP, Formatted select figures and edited the manuscript with input from all authors; Maria Victoria Neguembor, Data curation, Formal analysis, Methodology, Writing – review and editing, Performed and analysed STORM imaging; Manuel Beltran, Data curation, Formal analysis, Investigation, Methodology, Prepared cellular materials for CLIP, which was carried out by MB and MTC and supervised by RGJ; Wazeer Varsally, Formal analysis, Supervision, Methodology, Performed Hi-C, ChIP-seq and splicing analyses and supervised HP for some genomics approaches; Laura Martin, Data curation, Formal analysis, Methodology, Performed and analysed all STORM imaging; Manuel Tavares Cornejo, Data curation, Formal analysis, Methodology, Carried out CLIP with MB, supervised by RGJ; Dubravka Pezić, Conceptualization, Supervision, Methodology, Discovered splicing features of the SA isoforms and contributed conceptually to the study design; Amandeep Bhamra, Formal analysis, Investigation, Methodology, Performed mass spectrometric and proteomic analysis, supervised by SS; Silvia Surinova, Supervision, Performed and supervised mass spectrometry data analysis; Richard G Jenner, Supervision, Funding acquisition, Methodology, Writing – review and editing, Supervised CLIP; Maria Pia Cosma, Data curation, Funding acquisition, Investigation, Supervision, Writing – review and editing,

Performed and analysed STORM imaging; Suzana Hadjur, Conceptualization, Data curation, Funding acquisition, Methodology, Project administration, Supervision, Visualization, Writing – original draft, Writing – review and editing, Conceived the project, Obtained funding, Formatted all figures and wrote the manuscript with input from all authors

### Author ORCIDs
Hayley Porter http://orcid.org/0009-0004-1184-5583
Yang Li http://orcid.org/0000-0003-0943-6196
Maria Victoria Neguembor http://orcid.org/0000-0002-1583-1304
Manuel Beltran http://orcid.org/0000-0002-7945-2020
Laura Martin http://orcid.org/0000-0001-8801-6637
Dubravka Pezić http://orcid.org/0000-0002-9833-8469
Silvia Surinova http://orcid.org/0000-0003-0442-9595
Richard G Jenner http://orcid.org/0000-0002-2946-6811
Maria Pia Cosma http://orcid.org/0000-0003-4207-5097
Suzana Hadjur http://orcid.org/0000-0002-3146-3118

### Decision letter and Author response
Decision letter https://doi.org/10.7554/eLife.79386.sa1
Author response https://doi.org/10.7554/eLife.79386.sa2

## Additional files

### Supplementary files
• MDAR checklist

### Data availability
All data has been made freely available.

The following datasets were generated:

| Author(s) | Year | Dataset title | Dataset URL | Database and Identifier |
|---|---|---|---|---|
| Porter L, Hadjur S, Li Y | 2023 | Cohesin-independent STAG proteins interact with RNA and localise to R-loops to promote complex loading | https://www.ncbi.nlm.nih.gov/geo/query/acc.cgi?acc=GSE167887 | NCBI Gene Expression Omnibus, GSE167887 |
| Surinova S | 2023 | Identification of the protein interaction network of Stag proteins | https://www.ebi.ac.uk/pride/archive/projects/PXD024354 | PRIDE, PXD024354 |
| Surinova S | 2023 | Identification of the protein interaction network of STAG2 protein isoforms with inclusion and exclusion of exon32 | https://www.ebi.ac.uk/pride/archive/projects/PXD041975 | PRIDE, PXD041975 |

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
