## [Editor Report]

This study reports that the Stromalin Antigen (SA) proteins play a key role in the localization of the cohesin complex and promote loading of the complex. It shows that SA proteins interact with RNA, RNA binding proteins and R-loops, even in the absence of cohesin, providing evidence for a role for SA1 in cohesin loading which is independent of the canonical cohesin loader NIPBL. The study open new perspectives to understand the links between cohesin loading and chromatin structure that would rely on R-loops.

---

## [Decision Letter]

**Decision letter after peer review:**

Thank you for submitting your article "Cohesin-independent STAG proteins interact with RNA and localise to R-loops to promote complex loading" for consideration by *eLife*. Your article has been reviewed by 3 peer reviewers, one of whom is a member of our Board of Reviewing Editors, and the evaluation has been overseen by Kevin Struhl as the Senior Editor. The following individual involved in review of your submission has agreed to reveal their identity: Alessandro Vannini (Reviewer #2).

Essential revisions:

1) The R-loop IFs shown detect large spots of fluorescence that seem to mark the nucleolus. This needs to be clarified, using a nucleolus marker, since it would be possible that most of the effect related to R loops are occurring mainly at the rDNA regions

2) Authors should show by DRIP-qPCR and ChIP of SA at a number of regions that SA is enriched at the sites where R loops are enriched and that recruitment is diminished by RNH treatment. Similar experiments could be performed by using inhibitors of transcription such as DRB or some other.

3) The IF and global IP data shown in gels need to be quantified and measured properly with right controls. In this sense, authors seem to be concluding that the stability of SA proteins depends on its binding to RNA. This could be tested more specifically bby showing that after RNH treatment the protein disappear at higher speed whereas other proteins remain.

4) Provide experimental evidence of the existence of SA1 and/or SA2 bound to chromatin in isolation (ie without other components of the cohesin complex) in unperturbed cells. If data exists in the literature already, this can also be sufficient to infer their model. No evidence is provided of the existence of SA1 (or SA2) complexes bound to R-loops (or anywhere else) in normal circumstances (ie without induced degradation of RAD21). A Chip experiment (SA1 and/or SA2 vs RAD21) or co-localization imaging data would be at least supportive for the existence of SA1 and SA2 independent of the cohesin complex. Otherwise, we might be well just looking at remnants of what were cohesin holo-complexes after induced degradation of RAD21 subunits. Thus,

5) The analysis only considers exon 31/32 inclusion in HCT cells. It would be interesting to know whether the pattern seen here is consistent in other cell types, including non-cancer cell lines – this would indicate how relevant the specific SA1/2 differences seen here are to other cell types.

*Reviewer #1 (Recommendations for the authors):*

The study is certainly interesting. However, the conclusions need additional experimental support. It is known that cohesins are enriched at transcribed DNA regions and the association that the authors are really finding a role for SA protein during transcription and metabolism of nascent RNA, although the function is unclear, provided that this is mainly observed in the absence of cohesins. Authors provide data suggesting a role for SA protein in post-transcriptional gene expression, but this effect is investigated at that level. Stability of RNA, RNA export or some processing experiments are not provided. The association with DNA-RNA hybrid regions is weak, and needs more specific support. A major problem of this type of study is that once shown that a particular protein works in the context of chromatin and transcription, it is not surprising to find it interacting with nuclear factors involved in gene expression as well as with RNA and DNA.

– The association with R-loops needs further support. As such results can be interpreted in different manners. The IFs shown detect large spots of fluorescence that seems to mark the nucleolus. This needs to be clarified, using a nucleolus marker, since it would be possible that most of the effect related to R loops are occurring mainly at the rDNA regions, where it is known that hybrids accumulate at higher levels. If so, this would argue against the conclusions

– Authors need to show by DRIP-qPCR and ChIP of SA at a number of regions that SA is enriched at the sites where R loops are enriched and that recruitment is diminished by RNH treatment.

– Similar experiments should be performed by using inhibitors of transcription such as DRB or some others.

– The IF and global IP data shown in gels are not convincing, a part that could only reflect indirect effects. Data needs to be quantified and measured properly with right controls. Besides, authors seem to be concluding that the stability of SA proteins depends on its binding to RNA. This should be tested more specifically by kinetics after RNH treatment to show that indeed the protein evaporates at higher speed when treated with RNH whereas other proteins remain stable. This is important because RNH expression is toxic to cells and creates a stress situation with pleotropic consequences.

– Authors should draw a model.

– The large amount of data provided does not help the comprehension of the goals of the study. Authors should work better the writing by connecting in a rational way the different approaches undertaken

– Discussion could be improved. The amount of data provided are not sufficiently discussed to support the conclusions.

*Reviewer #3 (Recommendations for the authors):*

In order to show that cohesin loading occurs at R-loops, R-loop and cohesin ChIP after auxin wash off would be the ideal experiment. Ideally, this would include multiple timepoints and with/without SA depletion. However, this would be a significant amount of work and may be outside the scope of this work.

The analysis only considers exon 31/32 inclusion in HCT cells. It would be interesting to know whether the pattern seen here is consistent in other cell types, including non-cancer cell lines – this would indicate how relevant the specific SA1/2 differences seen here are to other cell types.

[Editors' note: further revisions were suggested prior to acceptance, as described below.]

Thank you for resubmitting your work entitled "Cohesin-independent STAG proteins interact with RNA and localise to R-loops to promote complex loading." for further consideration by *eLife*. Your revised article has been evaluated by Kevin Struhl (Senior Editor), a Reviewing Editor, and three reviewers.

We consider that the manuscript has been improved but there are some remaining issues that need to be addressed, as outlined below, before its final acceptance.

The authors have improved considerably the manuscript by clarifying a number of issues raised in the first revision. As a consequence of this, conclusions are supported in its large part. However, the main conclusion that SA has a direct functional role in R-loop regulation is not fully supported. Given the association of many proteins with transcribed regions, it is not unexpected that part of those regions may form R-loops, so an overlap of binding is observed. However, the fact that authors are not able to find a reduction by treating with RNH is worrying and provided that no biochemistry is added either. The study is certainly interesting, but conclusions in this regard should be lessened, particularly in the title and Abstract, and limitations regarding this point of the study raised in the Discussion.

---

## [Author Response]

Essential revisions:1) The R-loop IFs shown detect large spots of fluorescence that seem to mark the nucleolus. This needs to be clarified, using a nucleolus marker, since it would be possible that most of the effect related to R loops are occurring mainly at the rDNA regions

Based on existing literature, it is expected that a strong R-loop signal would be detected within the nucleolus. R-loops have been identified within the nucleolus by IF ^1,2^ and specifically at the rDNA locus using ChIP-qPCR ^3^. Indeed, our results recapitulate these observations, showing S9.6 signal within a Nucleolin mask by IF and S9.6 peaks within the intergenic spacer (IGS) of the rDNA consensus (Author response image 1, b).

**Author response image 1. sa2fig1:** (a) RAD21mAC cells stained for S9. 6 representing R-loops and Nucleolin. White line denotes the nucleus based on DAPI. NB. the expected nucleolar R-loop signal. (b) Top, cartoon of the consensus hg19 ribosomal DNA (rDNA), showing the ribosomal genes and the intergenic spacer (IGS) region which contains several Alu elements (blue). Bottom, S9.6 DRIP-seq from RAD21mAC cells treated with EtOH, IAA and RNaseH (RNH) and aligned to the consensus rDNA. NB. R-loops within the coding region are sensitive to RNH and IAA while those within the IGS are less affected. (C) Volcano plot displaying the statistical significance (-log10 p-value) versus magnitude of change (log2 fold change) from SA1 IPMS data produced from EtOH or IAA-treated RAD21mAC cells. Data as in Figure 2b from the manuscript except the proteins that are part of the enriched ‘Ribosome biogenesis’ functional category are highlighted in red.

We agree that some of the effect related to SA proteins and R-loops described here may indeed be occurring at rDNA. In fact, the literature supports this possibility. For example, it is known that cohesin is necessary for nucleolar integrity in yeast. Cohesin has been shown to bind to the nontranscribed region of the rDNA locus ^4^ and the 35S and 5S genes form loops that are dependent on Eco1, the cohesin subunit known to acetylate Smc3 and stabilize cohesin on chromatin ^5^. Consequently, yeast with eco1 mutations exhibit disorganised nucleolar structure and defective ribosome biogenesis. Similarly, we have discovered that Stag1 regulates nascent rDNA transcription and nucleolar integrity for mouse embryonic stem cell (mESC) pluripotency ^6^. This is likely to be direct since Stag1 is bound to both the mouse and human rDNA coding region and IGS, as well as at LINE and SINE repetitive elements important for nucleolar biology. Importantly, SA1/2 binding at rDNA and SINEs is maintained in IAA-treated RAD21^mAC^ cells and SA1^ΔCoh^ interacts with nucleolar proteins involved in rDNA transcription and processing, including Polr1a, UBF1 in the FC; Fibrillarin (FBL) in the DFC; and Nucleophosmin (NPM) and Nucleolin (NCL) in the GC (Review Figure 1c). Taken together, we argue that SA1 has an important and conserved role in maintaining nucleolar R-loops and structure.

2) Authors should show by DRIP-qPCR and ChIP of SA at a number of regions that SA is enriched at the sites where R loops are enriched and that recruitment is diminished by RNH treatment. Similar experiments could be performed by using inhibitors of transcription such as DRB or some other.

To further support our observations that SA proteins are localised to R-loops, we have prepared DRIP-seq in RAD21^mAC^ cells treated with RNase H (RNH), to confirm specificity of the DRIP signal, and with EtOH or IAA to assess the impact of cohesin loss on R-loops. We combined these datasets with our ChIP-seq for SA proteins and RAD21 in EtOH or IAA conditions. These results are discussed below.

Despite our best efforts, we were unable to address the question of whether SA1^ΔCoh^ is diminished by RNH treatment. First, RAD21^mAC^ cells experienced massive cell death after both IAA treatment and RNH overexpression. Next, we did manage to prepare SA ChIP-seq and DRIP-seq libraries upon triptolide treatment to inhibit transcription, however the ChIP signal/noise was extremely poor and we were not confident in these results. In the interest of time and resources, we proceeded with the analysis of R-loop and SA using DRIP-seq.

This revealed several interesting observations which have now been incorporated into the revised manuscript in a new layout (Lines 296-350, 1160-1186, Figure 3h, i, Figure 3 —figure supplement 1), and to our existing accession GSE167887. We confirmed the specificity of the experiment by identifying RNH-sensitive R-loops, among which we observed two regimes of SA association with R-loops. A small proportion of R-loop sites directly overlapped with SA1/2 in control conditions. These sites were enriched at genes and the SA signal was sensitive to RAD21 loss. On the other hand, a larger proportion of R-loops had SA signals adjacent (bound within 2kb of the R-loop peak). Interestingly, these SA sites were enriched in repressed chromatin and were not sensitive to RAD21 loss, in fact they were enriched.

Thus, our results support those already provided from IF, STORM, mass spec and coIP to show that cohesin and SA are localised to R-loops in control conditions, and that acute depletion of RAD21 leads to the continued associated of residual SA proteins at R-loops.

3) The IF and global IP data shown in gels need to be quantified and measured properly with right controls. In this sense, authors seem to be concluding that the stability of SA proteins depends on its binding to RNA. This could be tested more specifically bby showing that after RNH treatment the protein disappear at higher speed whereas other proteins remain.

Thank you for raising this point. While our results show that SA1^ΔCoh^ binds RNA and interacts with a number of RBP, we acknowledge that we do not definitively know that the stability of SA proteins (with or without cohesin) depends on its binding to RNA. We acknowledge that the treatment of RNases can impact global levels of many proteins and that the observation that SA1^ΔCoh^ is reduced upon RNH treatment is not sufficient in this context. Thus, we have removed the data from the manuscript and figures.

4) Provide experimental evidence of the existence of SA1 and/or SA2 bound to chromatin in isolation (ie without other components of the cohesin complex) in unperturbed cells. If data exists in the literature already, this can also be sufficient to infer their model. No evidence is provided of the existence of SA1 (or SA2) complexes bound to R-loops (or anywhere else) in normal circumstances (ie without induced degradation of RAD21). A Chip experiment (SA1 and/or SA2 vs RAD21) or co-localization imaging data would be at least supportive for the existence of SA1 and SA2 independent of the cohesin complex. Otherwise, we might be well just looking at remnants of what were cohesin holo-complexes after induced degradation of RAD21 subunits. Thus,

There are a number of pieces of evidence in the literature to support the existence of SA proteins on chromatin without core cohesin. These include a role for SA1 at telomeres ^7^; the effect of NIPBL loss on SA levels in hepatocytes (Figure 1) ^8^; and in vitro experiments of SAs ability to bind nucleic acids without cohesin ^9^.

To formally address this in our cell system, we have performed a GFP-TRAP in untreated (EtOH) RAD21^mAC^ cells. The results show efficient TRAP of RAD21, interactions with SA proteins as expected and ~20% ‘residual’ SA protein in the flow-through material after GFP-TRAP. We note that this amount of chromatin-bound SA which is not interacting with RAD21 in unperturbed cells is reminiscent of the amount of ‘residual’ chromatin-bound SA we observe upon acute degradation of RAD21. This result has been added to Figure 1 —figure supplement 1 and commented on at Lines 116-120 in the manuscript where we have also included Reference 7 above.

5) The analysis only considers exon 31/32 inclusion in HCT cells. It would be interesting to know whether the pattern seen here is consistent in other cell types, including non-cancer cell lines – this would indicate how relevant the specific SA1/2 differences seen here are to other cell types.

We have now included an additional analysis of exon 31/32 inclusion data for SA proteins from mouse embryonic stem (mESC) and mouse Neural progenitor cell (mNPC) RNA-seq datasets, including ones we have produced ourselves as well as published work. The data show that the splicing in of exon 31 of SA1 and the splicing out of exon 32 of SA2 is conserved across cell types and species, arguing for an important role for this splicing event in SA function in general. This data has been added to Figure 5 —figure supplement 1 and commented on at Line 456-457 in the manuscript.

References:

1. García-Rubio, M. L. *et al.* The Fanconi Anemia Pathway Protects Genome Integrity from R-loops. *PLoS Genet.* 11, e1005674 (2015).

2. Barroso, S. *et al.* The DNA damage response acts as a safeguard against harmful DNA-RNA hybrids of different origins. *EMBO Rep.* 20, e47250 (2019).

3. Abraham, K. J. *et al.* Nucleolar RNA polymerase II drives ribosome biogenesis. *Nature* 1–27 (2020). doi:10.1038/s41586-020-2497-0

4. Laloraya, S., Guacci, V. & Koshland, D. Chromosomal addresses of the cohesin component Mcd1p. *Journal of Cell Biology* 151, 1047–1056 (2000).

5. Harris, B. *et al.* Cohesion promotes nucleolar structure and function. *Mol Biol Cell* 25, 337–346 (2014).

6. Pezic, D. *et al. bioRxiv* 1–60 (2021). doi:10.1101/2021.02.14.429938

7. Bisht, K. K., Daniloski, Z. & Smith, S. SA1 binds directly to DNA through its unique AT-hook to promote sister chromatid cohesion at telomeres. *J. Cell. Sci.* 126, 3493–3503 (2013).

8. Schwarzer, W. *et al.* Two independent modes of chromatin organization revealed by cohesin removal. *Nature* 551, 51–56 (2017).

9. Pan, H. *et al.* Cohesin SA1 and SA2 are RNA binding proteins that localize to RNA containing regions on DNA. *Nucleic Acids Research* 24, 105–17 (2020).

[Editors' note: further revisions were suggested prior to acceptance, as described below.]

We consider that the manuscript has been improved but there are some remaining issues that need to be addressed, as outlined below, before its final acceptance.The authors have improved considerably the manuscript by clarifying a number of issues raised in the first revision. As a consequence of this, conclusions are supported in its large part. However, the main conclusion that SA has a direct functional role in R-loop regulation is not fully supported. Given the association of many proteins with transcribed regions, it is not unexpected that part of those regions may form R-loops, so an overlap of binding is observed. However, the fact that authors are not able to find a reduction by treating with RNH is worrying and provided that no biochemistry is added either. The study is certainly interesting, but conclusions in this regard should be lessened, particularly in the title and Abstract, and limitations regarding this point of the study raised in the Discussion.

As the SA ChIP in RNH conditions was not technically possible, we agree that we cannot definitively comment on whether SA has a role in R-loop regulation *per se*. We are however confident in the observation that cohesin-independent SA proteins interact with RBP and localise to R-loops in HCT cells. Thus, we have made adjustments to the Title, Abstract and Discussion to reflect this.

– The title has been changed from “Cohesin-independent STAG proteins interact with RNA and localise to R-loops to promote complex loading” to “Cohesin-independent STAG proteins interact with RNA and R-loops and promote complex loading”.

– We have also removed the comment that STAG proteins regulate R-loops from the abstract and replaced with “Accordingly, SA proteins interact with RNA, RNA binding proteins and R-loops, even in the absence of cohesin”.

Two comments in the Discussion have been changed:

1. Line 542 “…or perhaps sequentially, and facilitate the loading of cohesin. Future work investigating the binding profiles of SA proteins and re-loaded cohesin following R-loop depletion in vivo, would help to confirm these findings. Furthermore, biochemical assessment of SA interaction with R-loops will help identify the mechanisms of SA’s roles at R-loops both in the presence or absence of the cohesin ring”.

2. Line 569 “This study also reveals SA1 as a novel regulator of R-loop homeostasis” has been removed and replaced with “Our study suggests that SA1 may act as a novel regulator of R-loop homeostasis”.